# VAE-CycleGAN: Variational Latent Representation for Unpaired Image-to-Image Translation

## Abstract

Image-to-image translation plays a central role in computer vision, enabling applications such as style transfer, domain adaptation, and image enhancement. While recent advances have achieved strong paired translation results, learning mappings in unpaired settings remains challenging. In this work, we present a systematic comparison of autoencoder and variational autoencoder (VAE) variants for unpaired image-to-image translation, using paired data solely as a reference baseline. To capture distributional uncertainty, we introduce VAE-CycleGAN, a unified probabilistic framework that integrates variational inference into the CycleGAN architecture. Our method combines adversarial training and cycle-consistency with a VAE's probabilistic latent space, allowing the model to approximate the true posterior distribution. Further, the architecture achieves a $256\times$ spatial compression, efficiently compressing the input into a compact latent representation. Empirical results across the satellite-to-map benchmark dataset demonstrate that VAE-CycleGAN generates high-quality translated images (FID: 67.75) and achieves superior reconstruction fidelity (MSE: 0.0010, PSNR: 29.85 dB, SSIM: 0.7873) comparable to state-of-the-art deterministic approaches without hyperparameter tuning. For summer-to-winter and label-to-cityscape datasets, VAE-Cycle GAN performs comparably with state-of-the-art UNSB at 1 step, and is far superior to UNIT-DDPM at 1000 steps, while the determistic AE-CycleGAN is comparable to the 5-step USNB variant.

## 1 Introduction

Image-to-image translation, which involves mapping images from a source domain to a target domain while preserving content, has been widely studied in paired (Isola et al., 2017) and unpaired settings (Zhu et al., 2017a). While paired translation assumes aligned image pairs, datasets often lack such correspondence.

Applications like image design and artistic style transfer may have paired data, but scientific fields suffer from limited measurements, and unknown physical processes, making paired real-world or synthetic simulation data infeasible. Some examples include atmospheric remote sensing (satellite to maps, cloud cover, or vegetation), geophysical inversion (images of seismic wave velocity to natural resources), fluids (laminar to turbulent flow, images to velocity fields), and robotic vision (cityscape to labels). All these domains involve information asymmetry, leading to ill-posed translation.

Unpaired translation is thus more challenging. To this end, CycleGAN (Zhu et al., 2017a) introduced cycle-consistency for self-supervision. However, their deterministic mapping fails to capture the inherently multi-modal nature of many ill-posed translation tasks, where a single input corresponds to multiple outputs. Stochastic models have different problems: state-of-the-art diffusion models Ho et al. (2020) produce high fidelity images but have difficulty preserving positional information, with features slightly misaligned or shifted. Visually, these issues are negligible, but for scientific applications, computations such as mass, momentum, energy, area, volume can deteriorate, especially for chaotic simulations.

Meanwhile, nearly all state-of-the-art unpaired diffusion approaches rely either on coarse text-based manipulation or on a "translation module," typically implemented as an autoencoder. In practice,

achieving full posterior sampling requires semantic or latent noise, which in turn necessitates a pre-trained VAE. Thus, any high-performing unpaired translation model is implicitly built upon components closely aligned with the architecture we propose, even when not presented as such. Rather than posing a strict choice between VAEs and diffusion models, our findings indicate that a strong VAE is essential for building a strong diffusion model. We explored connecting the VAE directly to a diffusion process, following DiffuseVAE, and—even with only a tenth of the training—we observed more than a 6-point FID improvement over the base VAE's 65 FID.

Motivated by these insights and by the inherent multi-modality and positional sensitivity of scientific image-to-image translation tasks, we introduce VAE-CycleGAN, a unified framework that integrates the probabilistic latent space of a VAE into the CycleGAN architecture. Our model combines adversarial and cycle-consistent training with a structured latent distribution to enable diverse, controllable, and physically faithful translation. We further show that our deterministic variant, AE-CycleGAN, attains state-of-the-art fidelity—surpassing several U-Net and diffusion baselines, while preserving the precise spatial information crucial for scientific applications

## 2 RELATED WORK

Generative Adversarial Networks (GANs) (Goodfellow et al., 2014) introduce a transformative adversarial framework where a generator $(G)$ and discriminator $(D)$ are trained simultaneously through a minimax game. This approach produces high-fidelity samples efficiently via backpropagation, bypassing the need for Markov chains or explicit likelihoods. Subsequent developments improve training stability and sample quality (Radford et al., 2016; Arjovsky et al., 2017). Advancements on conditional GANs (cGANs) by Isola et al. (2017) enable directed generation but require paired data $(x, y)$. For high-resolution image generation, Wang et al. (2018) incorporates segmentation information, while Park et al. (2019) advances semantic manipulation capabilities. The challenge of multimodality in paired settings is addressed by Zhu et al. (2017b) with Bicycle-GAN, which combines cGANs with variational objectives to generate diverse outputs from single inputs.

However, paired data requirements render cGANs unsuitable for ill-posed inverse problems such as tomographic reconstruction or image colorization, where single inputs correspond to multiple valid outputs, as well as for distribution-level translation between unpaired domains like artistic style transfer. To address paired data limitations, unsupervised methods emerge. CycleGAN (Zhu et al., 2017a) and DualGAN (Yi et al., 2017) pioneer cycle-consistency loss with adversarial training for unpaired bi-directional mapping. UNIT (Liu et al., 2017) introduces shared-latent space assumptions, while StarGAN (Choi et al., 2018; 2020) enables multi-domain translation within a unified framework.

Subsequent efforts addressing deterministic limitations include MUNIT (Huang et al., 2018) and DRIT (Lee et al., 2018), which explicitly disentangle style and content codes for multimodal translation, albeit with increased architectural complexity. Meanwhile, Jha et al. (2018) combines a VAE with cycle-consistency, enforcing the reconstruction of an image after its latent factors (like style and content) have been translated across domains. Specifically, the latent space is disentangled into two complementary subspaces via weak supervision with pairwise similarity labels. Alternatively, Augmented CycleGAN (Almahairi et al., 2018) augments the latent space, enabling multimodality by cycling through different augmentations. Here, contrastive learning approaches as in Park et al. (2020) can leverage patch-wise losses between these augmented spaces.

Recently, flow-based models, including normalizing flows and diffusion models provide an alternative generative modeling paradigm through quasi-invertible transformations (Rezende & Mohamed, 2015; Kingma & Dhariwal, 2018), enabling both precise likelihood estimation and bi-directional latent space manipulation. Diffusion models (Ho et al., 2020; Song et al., 2021; Saharia et al., 2022; Zhang et al., 2023) utilize iterative denoising (stochastic) processes to achieve state-of-the-art performance in high-resolution image synthesis.

For unpaired translation, UNIT-DDPM (Sasaki et al., 2021) employs dual denoising diffusion models trained jointly via conditional score-matching. They find improved Fréchet Inception Distance (FID) performance over prior approches like CycleGAN and VAEs, though later retraining shows similar CycleGAN performance (Kim et al., 2024).

EGSDE (Zhao et al., 2022) models image translation through energy-guided stochastic differential equations (SDEs), offering enhanced training stability and stochasticity not present in CycleGAN. Similarly, Unpaired Image-to-Image Translation via Neural Schrödinger Bridge (NSB) (Sasaki et al., 2021) frames translation as continuous-time stochastic interpolation. Both of these approaches forgo an explicit, interpretable latent space for higher visual fidelity (FID), and generally outperform CycleGAN. As we show, this need not be the case; explicit, stochastic latent space autoencoders can provide near-equivalent performance with better architecture.

In particular, modern VAE architectures have substantially advanced beyond the original formulation (Kingma et al., 2013). Hierarchical VAEs (Vahdat & Kautz, 2020) employ multi-scale latent representations to capture complex data distributions. Vector-quantized VAEs (VQ-VAE) (Van den Oord et al., 2017) introduce discrete latent representations through a codebook mechanism, effectively circumventing the posterior collapse problem common in continuous VAEs when paired with powerful decoders. This approach replaces the continuous latent space with discrete codes learned via vector quantization, creating a robust framework for high-quality image, video, and speech generation. The subsequent VQ-VAE-2 (Razavi et al., 2019) enhances this foundation through a multi-scale hierarchical architecture, employing powerful autoregressive priors over the latent codes to generate high-fidelity, diverse samples that rival state-of-the-art GANs, while maintaining the training stability and diversity advantages of VAE-based approaches.The NVAE architecture (Vahdat & Kautz, 2020) uses depth-wise separable convolutions to also demonstrate that hierarchical VAEs can compete with other state-of-the-art generative models; the Very Deep VAE (Child, 2021) shows that extremely deep hierarchical variational models can rival autoregressive approaches in sample quality.

Finally, the VAE-GAN paradigm (Larsen et al., 2016) combines variational autoencoders' latent space learning with GANs' adversarial training, using the discriminator for perceptually-aware reconstruction losses rather than pixel-wise metrics. This hybrid approach demonstrates enhanced generalization and output fidelity (Yan et al., 2025; Denton & Fergus, 2019), with the VAE learning meaningful latent representations while the adversarial component ensures distributional alignment.

Our proposed VAE-CycleGAN builds upon these advances, incorporating cycle-consistent adversarial training for bi-directional unpaired translation while leveraging VAE-based generators to introduce multimodality and structured latent spaces. This distinguishes our work from standard VAE-GANs (Larsen et al., 2016; Yan et al., 2025) and addresses CycleGAN's determinism through intrinsic stochasticity rather than external auxiliary variables (Almahairi et al., 2018). While we do not explicitly include diffusion models, our VAE-based model allows for easy integration with latent diffusion models as in Zhang et al. (2023) or refinements like Pandey et al. (2022).

Concurrent research by Sharma et al. (2025) explored medical image translation with unpaired data using a similar combination of a VAE and a CycleGAN. Their architecture employs two GANs where the generator module of only one GAN incorporates a VAE neural network. In contrast, our network integrates a VAE into each of the two GANs, fully leveraging the potential for variability in the translated images. Furthermore, our paper provides a comprehensive comparison of different AE (autoencoder) and VAE variants in terms of reconstruction, translation, and diversity of the translated samples, with both perception (visual quality) and distortion (pointwise accuracy) metrics.

## 3 PROBLEM FORMULATION

In consistent image translation, we expect a canonical isomorphism between two visual domains $X, Y$. In practice, all possible such mappings are lossy and thus non-invertible; we thus seek the maximally structure-preserving map (or pair of homomorphisms with minimal kernel). These maps are not unique; given an image $x_i \in X$, we can define the possible outcomes $\hat{y} \sim p_{\hat{Y}|X}$ with a posterior distribution, as in classical ill-posed inverse problems.

As in CycleGAN (Zhu et al., 2017a), we indirectly enforce maximal structure preservation through a *cycle consistency loss*, for generators $G : X \to Y, F : Y \to X$.

$$\mathcal{L}_{\text{cycle}} = \mathbb{E}_x \left[ |x - F(G(x))|_1 \right] + \mathbb{E}_y \left[ |y - G(F(y))|_1 \right] \tag{1}$$

Since this is a distortion metric, our model follows the perception-distortion tradeoff (Blau & Michaeli, 2018): an estimator cannot simultaneously achieve optimal accuracy (minimal distortion) and optimal perception (distributional or visual quality). For an allowable distortion level $D$,

perception metric (f-divergence, e.g. Kullback-Leibler (KL)) $d$, and distortion metric (e.g. L1 loss) $\Delta$, the optimal model satisfies Eqn. 2:

$$\hat{p}_{\hat{y}|x} = \underset{p_{\hat{y}|x}}{\arg\min} \, d(p_y, p_{\hat{y}}) \quad \text{s.t.} \quad \mathbb{E}_{x,y \sim p_{\text{data}}} \mathbb{E}_{\hat{y} \sim p_{\hat{y}|x}} \left[ \Delta(y, \hat{y}) \right] \leq D \tag{2}$$

Low distortion estimators (e.g., standard VAEs, autoencoders) minimize $\mathbb{E}[\Delta(y, \hat{y})]$ but ignore the distribution $p_{\hat{y}}$, resulting in blurry, perceptually unrealistic outputs (converging to the pixel-wise maximum a-posteriori solution). In contrast, high perception estimators (e.g., GANs) learn and minimize $d(p_y, p_{\hat{y}})$ and produce sharp, realistic samples but provide no guarantee that a generated sample $\hat{y}$ is a faithful reconstruction of a specific input $x$. An optimal inversion requires sampling from the full posterior $p(y \mid x)$, which defines the Pareto frontier in Eqn. 2. VAE-CycleGAN is designed to learn this posterior distribution, enabling both perceptually realistic and distortion-faithful reconstructions. We use for perception metric $d(p_y, p_{\hat{y}})$ an *adversarial $\chi^2$ divergence* between generated images and target domains, with discriminators $D_X : X \to \mathbb{P}_X, D_Y : Y \to \mathbb{P}_Y$.

$$\mathcal{L}_{\text{GAN}}^{X \to Y} = \mathbb{E}_y[D_Y(y)^2] + \mathbb{E}_x[(1 - D_Y(G(x)))^2] \tag{3}$$

$$\mathcal{L}_{\text{GAN}}^{Y \to X} = \mathbb{E}_x[D_X(x)^2] + \mathbb{E}_y[(1 - D_X(F(y)))^2] \tag{4}$$

Interestingly, at no point have we required paired data; by quantifying both perception and distortion the model is now fully specified, even in the unpaired setting (where we have datasets: $\{(x_i \in X) \sim p_{\text{data}}(x)\}_{i=1}^N$ and $\{(y_j \in Y) \sim p_{\text{data}}(y)\}_{j=1}^M$). Please see Appendix 7.1 for a proof sketch of model convergence to a natural map, though not necessarily a human-preferred, canonical map.

We finish the objectives with regularizers. For fast posterior sampling, a *$\beta$-VAE loss* regularizes the latent space to a normal distribution with a KL divergence $d_{KL}$ (Higgins et al., 2017),(Weng, 2018). For latent variable $z$, $\phi$ and $\theta$ parameterize the encoder $q_\phi(z|\cdot)$ and decoder $p_\theta(\cdot|z)$ respectively. For stability, we add an *identity loss* regularizer to preserve content during adversarial training.

$$\mathcal{L}_{\text{VAE}}^X = \mathbb{E}_{z \sim q_\phi(z|x)} \, d_{\text{KL}}\big(q_\phi(z|x) \,\|\, p(z) \triangleq \mathcal{N}(0, \mathbb{I})\big) \tag{5}$$

$$\mathcal{L}_{\text{VAE}}^Y = \mathbb{E}_{z \sim q_\phi(z|y)} \, d_{\text{KL}}\big(q_\phi(z|y) \,\|\, p(z) \triangleq \mathcal{N}(0, \mathbb{I})\big) \tag{6}$$

$$\mathcal{L}_{\text{identity}} = \mathbb{E}_x \left[ |G(x) - x|_1 \right] + \mathbb{E}_y \left[ |F(y) - y|_1 \right] \tag{7}$$

The complete training objectives are then as in Eqns. 8, 9, where $\lambda_{\text{GAN}}$, $\lambda_{\text{cycle}}$, $\lambda_{\text{id}}$, $\lambda_{\text{kl}} << 1$ are weighting coefficients.

$$\text{AE-CycleGAN: } \mathcal{L}_{\text{Total}} = \lambda_{\text{GAN}} \mathcal{L}_{\text{GAN}} + \lambda_{\text{cycle}} \mathcal{L}_{\text{cycle}} + \lambda_{\text{id}} \mathcal{L}_{\text{identity}} \tag{8}$$

$$\text{VAE-CycleGAN: } \mathcal{L}_{\text{Total}} = \lambda_{\text{kl}} \mathcal{L}_{\text{VAE}} + \lambda_{\text{GAN}} \mathcal{L}_{\text{GAN}} + \lambda_{\text{cycle}} \mathcal{L}_{\text{cycle}} + \lambda_{\text{id}} \mathcal{L}_{\text{identity}} \tag{9}$$

## 3.1 NETWORKS

We train the following variants in Table 1 to fully ablate these objectives, on an asymmetric translation benchmark between high-resolution satellite (aerial) photos $X$ and simplified map-like representations $Y$, with comprehensive architecture as in Figure 1. We closely follow the CycleGAN architecture, but remove the U-Net skip connections and increase compression by 12x. Upsampling and downsampling instead use pixel shuffle-unshuffle as in Chen et al. (2024). Please see Appendix 7.2, 7.3 for exact implementation details, respectively.

Table 1: Architecture variants

| | Model | Type | Objectives |
|---|---|---|---|
| **Paired Dataset** | AE | Deterministic | Translation |
| | Cycle AE | Deterministic | Translation, Cycle consistency |
| | AE-GAN | Deterministic | Translation, Adversarial, Identity |
| | VAE | Stochastic | Translation, KL-Divergence |
| | Cycle-VAE | Stochastic | Translation, Cycle consistency, KL-Divergence |
| | VAE-GAN | Stochastic | Translation, Adversarial, Identity, KL-Divergence |
| **Unpaired** | Cycle AE | Deterministic | Cycle consistency |
| | AE-CycleGAN | Deterministic | Adversarial, Identity, Cycle consistency |
| | Cycle-VAE | Stochastic | Cycle consistency, KL-Divergence |
| | VAE-CycleGAN | Stochastic | Adversarial, Identity, Cycle consistency, KL-Divergence |

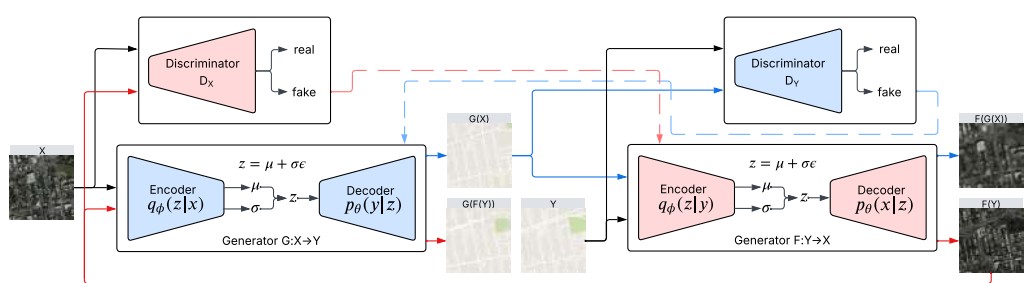

Figure 1: VAE-CycleGAN for unpaired translation. Generators $G$ and $F$ map $X \to Y$ and $Y \to X$, respectively, discriminators $D_X$ and $D_Y$ enforce domain alignment, and reconstruction losses $F(G(X)) \approx X$ & $G(F(Y)) \approx Y$ enforce cycle-consistency.

## 4 RESULTS

We first compare the performance of paired dataset AE and VAE variants, then extend to unpaired settings. We then return to the role of adversarial loss and cycle-consistency in the stochastic (inverse-problem) case, by introducing a Gaussian prior in the latent space.

To evaluate performance, we use distortion-based metrics MSE, PSNR, the perception-oriented Structural Similarity Index (SSIM), and perception-based metrics including Fréchet Inception Distance (FID) and Kernel Inception Distance (KID), to measure distributional distance between real and generated images. We report KID scores since our datasets are relatively small with only about 1K training and testing samples.

Note that all our AE/VAE variants are quite stable and rarely collapses; we see almost no mode collapse provided that the loss weights are not extremely unbalanced. As such, we do not / have not finetuned the loss weights and other hyperparameters beyond scaling the cycle consistency to adequately preserve information.

For brevity, we present an examples of VAE-CycleGAN's translation and reconstruction (unpaired) in Figures 2, 3. Examples of all AE/VAE variants (listed in Table 1) are presented in Appendix 7.4.

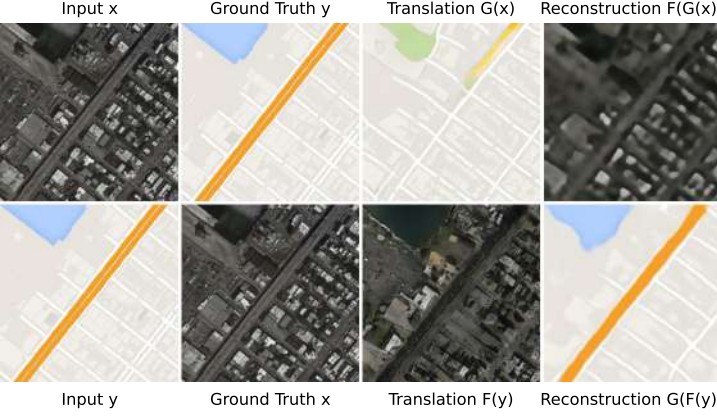

Figure 2: Satellite images $\leftrightarrow$ Maps: VAE-CycleGAN translated and reconstructed outputs.

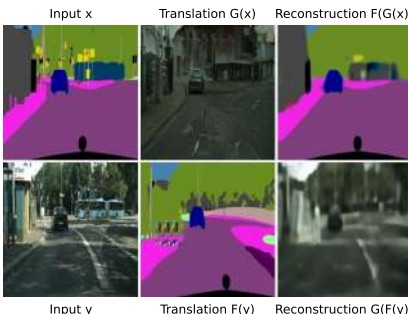

Figure 3: Summer $\leftrightarrow$ Winter, Labels $\leftrightarrow$ Cityscape: VAE-CycleGAN translated and reconstructed outputs. Translations are sharp with the exception of the Winter to Summer case, as the input $y$ could be easily classified as either summer or winter.

Tables 2 and 3 display the average translation and reconstruction errors for the Satellite2Map dataset.

Table 2: Model ablation across translation tasks. KID is scaled by 100.

| | Model | $G : X \to Y$ (aerial→map) | | | | | $F : Y \to X$ (map→aerial) | | | | |
|---|---|---|---|---|---|---|---|---|---|---|---|
| | | MSE ↓ | PSNR ↑ | SSIM ↑ | FID ↓ | KID $\mu \pm \sigma$ ↓ | MSE ↓ | PSNR ↑ | SSIM ↑ | FID ↓ | KID $\mu \pm \sigma$ ↓ |
| **Paired** | AE | **0.0018** | 27.37 | 0.7636 | 325.56 | 35.30 ± 3.74 | 0.0256 | 14.08 | 0.2489 | 253.05 | 23.31 ± 1.96 |
| | Cycle AE | **0.0018** | **27.46** | **0.7650** | 273.17 | 27.29 ± 2.41 | **0.0251** | 14.13 | **0.2553** | 241.77 | 21.75 ± 1.89 |
| | AE-GAN | 0.0031 | 25.10 | 0.6684 | **63.83** | **3.16** ± 0.66 | 0.0349 | **14.29** | 0.2079 | **52.33** | **1.75** ± 0.55 |
| | VAE | 0.0024 | 26.03 | 0.6753 | 358.70 | 35.47 ± 3.10 | 0.0261 | 13.50 | 0.1894 | 304.34 | 28.58 ± 2.09 |
| | Cycle VAE | 0.0022 | 26.58 | 0.7117 | 325.89 | 33.69 ± 2.36 | 0.0255 | 13.25 | 0.2080 | 259.83 | 24.03 ± 1.89 |
| | VAE-GAN | 0.0033 | 24.85 | 0.6272 | 83.18 | 5.10 ± 0.91 | 0.0354 | 14.17 | 0.1705 | 64.45 | 3.01 ± 0.52 |
| **Unpaired** | Cycle AE | 0.0872 | 10.59 | 0.0318 | 409.27 | 50.77 ± 1.63 | 0.0410 | **13.69** | 0.0924 | 329.51 | 35.52 ± 1.75 |
| | Cycle VAE | 0.0113 | 19.47 | 0.3128 | 419.51 | 53.93 ± 1.94 | 0.0614 | 7.90 | 0.0885 | 475.98 | 59.31 ± 2.74 |
| | AE-CycleGAN | **0.0050** | **22.97** | **0.6737** | **70.08** | **4.60** ± 1.31 | **0.0389** | 13.62 | **0.1641** | **52.53** | **1.70** ± 0.48 |
| | VAE-CycleGAN | 0.0056 | 22.50 | 0.5793 | 90.87 | 5.94 ± 0.92 | 0.0443 | 13.36 | 0.0965 | 69.25 | 3.78 ± 0.63 |

Table 3: Model ablation across reconstruction tasks. KID is scaled by 100.

| | Model | $G(F(y))$: Map Reconstruction | | | | | $F(G(x))$: Aerial Reconstruction | | | | |
|---|---|---|---|---|---|---|---|---|---|---|---|
| | | MSE ↓ | PSNR ↑ | SSIM ↑ | FID ↓ | KID $\mu \pm \sigma$ ↓ | MSE ↓ | PSNR ↑ | SSIM ↑ | FID ↓ | KID $\mu \pm \sigma$ ↓ |
| **Paired** | AE | 0.0017 | 27.77 | 0.7718 | 315.15 | 33.47 ± 3.24 | 0.0287 | 13.95 | 0.1992 | 288.35 | 28.07 ± 2.08 |
| | Cycle AE | **0.0003** | **35.91** | **0.9242** | 83.08 | 6.04 ± 0.66 | **0.0059** | **21.53** | **0.6661** | 175.29 | 16.21 ± 1.64 |
| | AE-GAN | 0.0050 | 22.99 | 0.6969 | **74.76** | **4.25** ± 0.82 | 0.0392 | 14.29 | 0.1481 | **70.75** | **3.47** ± 0.67 |
| | VAE | 0.0049 | 22.63 | 0.6717 | 367.13 | 38.95 ± 2.50 | 0.0340 | 11.13 | 0.1344 | 347.17 | 36.42 ± 1.88 |
| | Cycle VAE | 0.0007 | 31.81 | 0.8450 | 170.27 | 15.46 ± 1.54 | 0.0145 | 17.03 | 0.3330 | 266.76 | 25.87 ± 1.73 |
| | VAE-GAN | 0.0060 | 22.22 | 0.6172 | 104.49 | 8.00 ± 1.07 | 0.0430 | 13.35 | 0.0954 | 84.63 | 4.96 ± 1.01 |
| **Unpaired** | Cycle AE | **0.0002** | **36.59** | **0.9317** | **80.05** | **6.08** ± 0.75 | **0.0063** | **21.12** | **0.6451** | **180.12** | **16.62** ± 1.88 |
| | Cycle VAE | 0.0006 | 32.44 | 0.8468 | 219.37 | 21.30 ± 2.82 | 0.0146 | 17.31 | 0.3151 | 281.64 | 27.61 ± 1.68 |
| | AE-CycleGAN | 0.0005 | 32.99 | 0.8760 | 106.63 | 8.44 ± 0.92 | 0.0096 | 19.01 | 0.5069 | 200.11 | 18.33 ± 1.81 |
| | VAE-CycleGAN | 0.0011 | 29.67 | 0.7804 | 241.78 | 24.09 ± 2.59 | 0.0175 | 16.10 | 0.2779 | 271.72 | 27.03 ± 2.00 |

Table 4: Finetuned translation tasks. The AE-CycleGAN is nearly state-of-the-art.

| Model | FID ↓ | Steps | Label2Cityscape $\to$ | Summer2Winter $\to$ | $\leftarrow$ | Satellite2Map $\to$ | $\leftarrow$ |
|---|---|---|---|---|---|---|---|
| UNIT-DDPM Sasaki et al. (2021) | 1000 | 113.70 | 109.98 | 113.70 | 116.23 | 193.06 | |
| MUNIT Huang et al. (2018) | 1 | 91.4 | 115.4 | - | - | 181.7 | |
| CycleGAN Zhu et al. (2017a) | 1 | 76.3 | 84.9 | - | - | 54.6 | |
| CycleDiff Zou et al. (2025) | 100 | **45.1** | **72.7** | - | - | 53.2 | |
| UNSB Kim et al. (2024) | 1 | ≈ 74 | ≈ 75 | - | - | ≈ 71 | |
| UNSB Kim et al. (2024) | 5 | 53.2 | 73.9 | - | - | **47.6** | |
| AE-CycleGAN | 1 | - | - | - | **71.1** | 49.4 | |
| VAE-CycleGAN (lower $\lambda_{kl}$) | 1 | - | - | - | 89.8 | 57.3 | |
| VAE-CycleGAN | 1 | 67.9 | 86.6 | **72.4** | 98.4 | 67.7 | |

The asymmetric difficulty of the aerial and map datasets is immediately obvious, with any aerial task generally worse in distortion metrics (high MSE, low PSNR, SSIM) but easier to capture in-distribution (low FID, KID). Regardless, optimal translation and reconstruction models closely follow the perception-distortion trade-off (Blau & Michaeli, 2018). In both paired translation and reconstruction, Cycle-AE minimizes distortion via a cycle-consistency loss (lowest MSE, high PSNR, SSIM), while AE-GAN optimizes for perception with the adversarial loss (lowest FID, KID). Interestingly, in unpaired translation, the adversarial loss becomes crucial for domain alignment. Consequently, AE-CycleGAN achieves the best overall performance, superior in both distortion and perception metrics. Similarly, Cycle AE (with only cycle-consistency) performs best in all reconstruction metrics, as cycle-consistency is crucial for information preservation.

We find AE-CycleGAN the best overall performing (deterministic) model. No metric in Table 2 directly quantifies the conditional posterior (distribution of cycle-consistent translations), since that is intractable over the large 256x256x3 image space, so stochastic models are expected to perform worse on this benchmark. FID and KID only quantify the marginal posterior (any conditioned translation regardless of consistency). Nevertheless, unlike the other stochastic VAE models, VAE-CycleGAN performs closely behind AE-CycleGAN in translation. We expand on this with qualitative measurements in section 5.

Against other state-of-the-art U-Net and diffusion models, AE-CycleGAN achieves comparable fidelity. We outperform the CycleGAN U-Net (Zhu et al., 2017a) at 12x (greater) compression by simply modernizing the autoencoder portion: we remove skip connections and switch to pixel shuffle layers. We also outperform MUNIT (U-Net), UNIT-DDPM and CycleDiff (U-Net with multiple steps). Finally, we outperform the state-of-the-art UNSB (Kim et al., 2024) at 1-step and are comparable with their 5-step model. No comparisons are made against text-conditioned diffusion models; those models are neither quantitative enough for scientific applications, nor are they pixel-level cycle consistent. Also, the VAE-CycleGAN cannot be directly compared with these diffusion models, as state-of-the-art diffusion models are built on top of VAEs; by design, diffusion models need not approximate the complete posterior given limited training data (which is why conditional models and posterior sampling were developed). In some cases, the models actually memorize the training data (Somepalli et al., 2023).

We now visualize translations from aerial photos to maps, $x \to G(x)$ in Figure 4 and the aerial image reconstructions, $x \approx F(G(x))$ in Figure 5 for additional qualitative comparison.

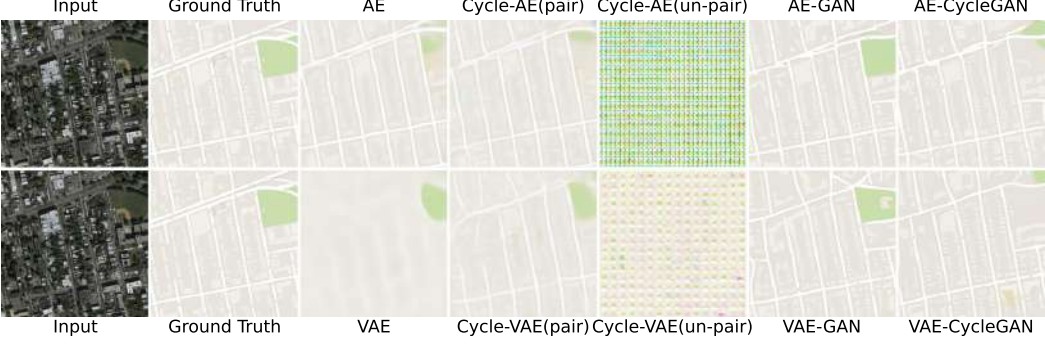

Figure 4: Left to right: input $x$, target $y$, map translations $G(x)$. Top to bottom: AE/VAE variants.

Per Figure 4, maps translated by the Cycle-AE and Cycle-VAE models in an unpaired data setting show no structural similarity to the map domain, as there is neither an adversarial constraint nor a paired dataset to ensure domain alignment. However, spatial information is preserved and allows for the complete reconstruction of the original aerial input, as shown in Figure 5. We notice a similar pattern for aerial photos translation, $y \to F(y)$ in Figure 6 and map reconstruction, $y \approx G(F(y))$ in Figure 7.

Visually, VAE-CycleGAN and AE-CycleGAN both produce the highest quality translations, with the VAE-CycleGAN enabling distributional sampling at a moderate hit to reconstruction quality, as expected from the perception-distortion tradeoff Blau & Michaeli (2018).

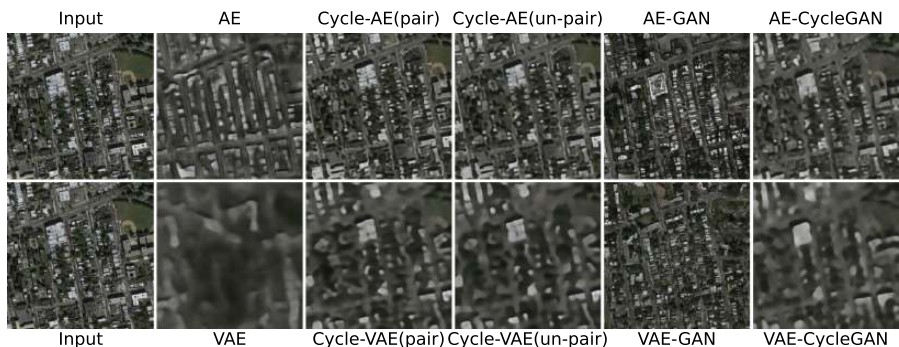

Figure 5: Left to right: input $x$, aerial reconstructions $F(G(x))$. Top to bottom: AE/VAE variants.

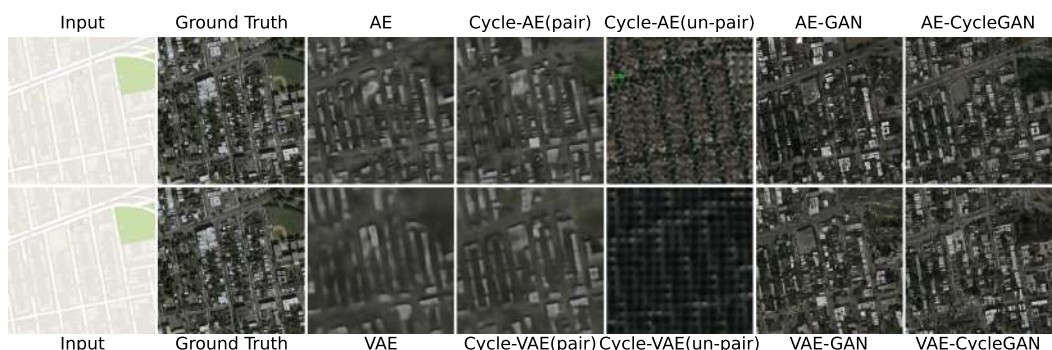

Figure 6: Left to right: input $y$, target $x$, aerial translations $F(y)$. Top to bottom: AE/VAE variants.

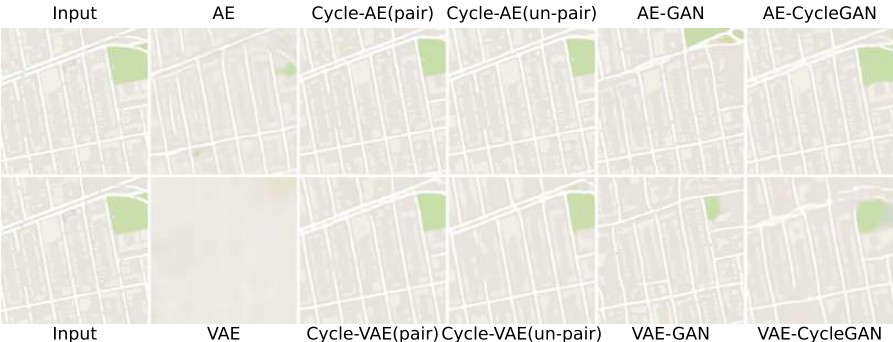

Figure 7: Left to right: input $y$ and map reconstructions $G(F(y))$. Top to bottom: AE/VAE variants.

## 5 REALIZATIONS

Realizations, ensemble mean, standard deviation, and ensemble error together form a qualitative measure of the intractable conditional performance mentioned in section 4. For brevity, we now visualize VAE-CycleGAN realizations and ensemble statistics in Figures 8, 9. Please find expanded figures and metrics in Appendix 7.5 and 7.6; figures 54 & 53 and 51 & 52 show translations/reconstructions for the cityscape-to-label dataset and summer-to-winter dataset.

As expected, we find a tradeoff in feature sharpness. In aerial output from Figure 8, we find homogeneous, low-information areas (such as roads and open fields) lose sharpness, while sharp edges and textured areas (such as buildings) gain sharpness. Similar behavior is visible in the ensemble mean (Figure 9). Cycle-consistency is satisfied on the low-information areas (even if the translation

Figure 8: VAE-CycleGAN sample realizations. Row 1: aerial$\to$ maps, row 2: maps $\to$ aerial

is missing information, the details are trivially reconstructed), so it follows that the adversarial loss is poorly aligned with human perception.

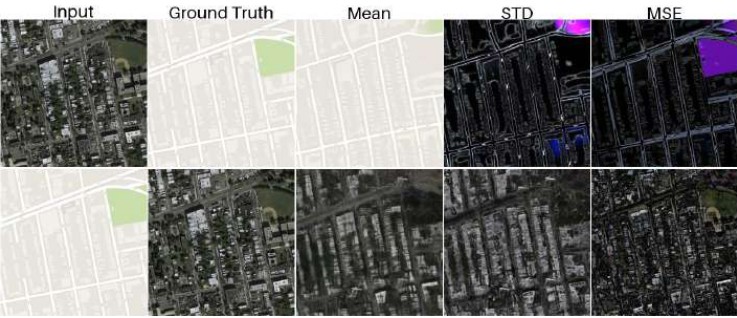

Figure 9: Mean, scaled STD (aerial $3\times$, maps $20\times$), and MSE across 30 realizations. The standard deviation and MSE are reported in true color, with purple color in standard deviation demonstrating that the generator is confident in green value but highly variable in red and blue. Such behavior is especially prominent in heavily treed or grassy areas (green areas) in satellite images; as expected, the generators learn green value well due to the imbalanced dataset (trees and grass dominate the satellite images).

As no VAE variant can outperform the best deterministic autoencoder, from the perception-distortion tradeoff, we instead follow standard Bayesian modeling skill. Skill is defined as the correlation of ensemble mean error and standard deviation, and we find that across realizations, the model does produce diverse outputs for regions of uncertainty. In satellite to map translation, uncertainty is highest when translating low-detail ground features (like grass or empty fields) as there is simply not enough information for accurate translation. In map to satellite translation, high-detail textures (like buildings) naturally have highest uncertainty (as an ill-posed inverse problem). Across an ensemble of 30 samples, the map translation MSE is $\approx 6.4 \times 10^{-3}$ while the aerial translation MSE is $\approx 35.1 \times 10^{-3}$ for VAE-CycleGAN, due to the greater ill-posedness of the latter.

For all VAE models, Figures 10 and 11 display ensemble summaries. For visibility, map standard deviation images for all networks were scaled by a factor of 20 (excepting VAE-CycleGAN at $25\times$) and then clamped to the 0-1 range for display. Similarly, aerial standard deviation images were scaled by $10\times$ for VAE and Cycle VAE variants, by $3\times$ for GAN based VAEs, and also clamped.

In Figures 10 and 11, we observe that the ensemble means for the GAN-based VAEs are much sharper than the VAE and paired Cycle-VAE, despite the paired data. Conversely, the unpaired Cycle-VAE completely lacks domain alignment, without the adversarial loss or paired data. For nearly all models except VAE and the unpaired Cycle-VAE, standard deviation is again a good estimator of ensemble MSE. Lastly, note VAE-CycleGAN shows some positional distortion in the map due to the limited domain alignment possible in the unpaired setting; it is the only performant unpaired stochastic VAE model.

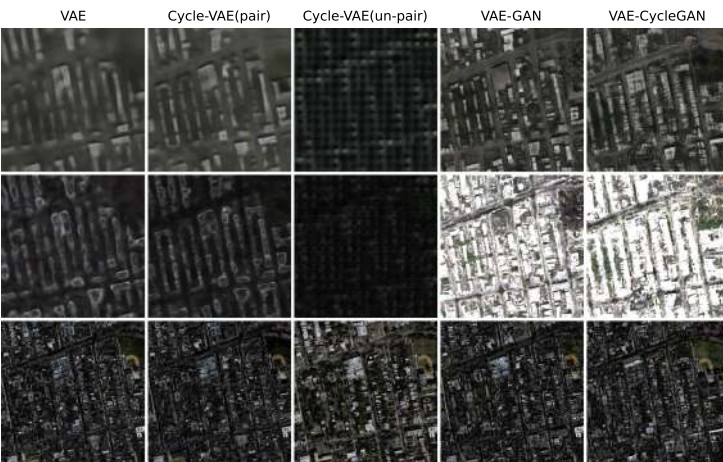

Figure 10: Rows: mean, STD (scaled for visibility), and MSE of 30 aerial realizations (VAE variants).

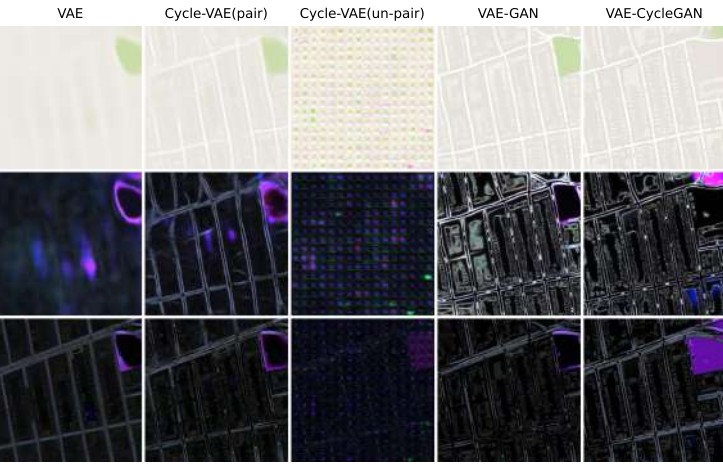

Figure 11: Rows: mean, STD (scaled for visibility), and MSE of 30 map realizations (VAE variants).

# 6 CONCLUSION

The AE-CycleGAN and VAE-CycleGAN models perform competitively both on distortion and perception metrics during translation. VAE-CycleGAN further maintains output diversity, at an acceptable fidelity tradeoff. The variational latent space allows future adaptation with multi-step methods such as diffusion models, enabling fine-grained or prompt-based control.

As our framework applies variational inference and modern autoencoders to the CycleGAN architecture, extensions to general class of unpaired, bidirectional ill-posed inverse problems are straightforward. Possible applications beyond natural images could include molecular design and medical image synthesis (CT-to-MRI synthesis, PET-to-CT conversion, etc.) from 3D or other non-image data.

ACKNOWLEDGMENTS

The authors acknowledge the use of AI tools for visualization code and language refinement for clarity. All AI-generated content was critically reviewed, adapted, and validated to ensure scientific accuracy, with the authors maintaining full responsibility for the research and intellectual content.

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

# 7 APPENDIX

## 7.1 CONVERGENCE AND OPTIMALITY

To our knowledge, Zhu et al. (2017a) do not provide a proof of convergence for a cycle-consistent framework. We provide a quick sketch below.

We begin by defining the spaces and mappings relevant to the cycle-consistent neural network framework. Let $X$ denote the input domain, where each element $x \in X$ represents one of $n$ distinct items (e.g., images), as determined by the dimensionality of the input data. Let $Y$ be the target output domain, where each $y \in Y$ likewise corresponds to one of $n$ distinct items, consistent with the output data dimension. Let $f : X \to Z$ denote the forward mapping implemented by the neural network, where $Z$ is an intermediate representation space. Let $g : Z \to X$ be the inverse mapping used to reconstruct the original input from the network's output. In the context of a model trained with cycle-consistency loss, we assume the following two properties:

$$\forall x \in X, \exists g \text{ such that } g(f(x)) = x \tag{10}$$

$$Z = Y \tag{11}$$

Equation 10 ensures the existence of a cycle-consistent mapping. Equation 11 states that the intermediate representation space $Z$ is equivalent to the target output domain $Y$, thereby implying that the network effectively learns a mapping from $X$ to $Y$.

Given that $f$ is bijective (by Equation 10) and that $|X| = |Y| = n$ (by Equation 11), it follows that there exist $n!$ possible one-to-one mappings (i.e., permutations) between elements of $X$ and $Y$. Although many such bijective mappings exist in theory, the cycle-consistency loss biases the network toward converging on a single, consistent, and invertible transformation that minimizes the reconstruction error.

Let $f_\theta$ denote the forward neural network (parameterized by weights $\theta$) which maps inputs from domain $X$ to outputs in domain $Y$. Let $g_\theta$ denote the inverse neural network, also parameterized by $\theta$, that attempts to reconstruct the original input from the output of $f_\theta$. By Equation 10 and $|X| = |Y| = n$, note that $g = f^{-1}$. The cycle-consistency loss $\mathcal{L}(\theta)$ is then defined as the expected reconstruction error between the original input $x$ and its reconstruction $g_\theta(f_\theta(x))$, measured using the squared $L^2$ norm:

$$\mathcal{L}(\theta) = \mathbb{E}_{x \sim X} \left[ \|g_\theta(f_\theta(x)) - x\|_2^2 \right] \tag{12}$$

The optimal parameters $\theta^*$ are obtained by minimizing the cycle-consistency loss: $\theta^* = \arg\min_\theta \mathcal{L}(\theta)$.

In general, we assume the following about the solution ($\theta^*$) landscape:

1. No local minima exist (i.e., network optimizer will never be stuck at a local minima)
2. There exists a unique $\theta^*$ such that $\mathcal{L}(\theta^*) < \epsilon$ for some $\epsilon \in \mathbb{R}^+$

Solution uniqueness is enforced by the neural network's inherent incompleteness: the neural network cannot perfectly reconstruct $x$, i.e., $\mathcal{L}(\theta) > 0 \ \forall \theta$. Since exact recovery is impossible, the model cannot satisfy cycle-consistency for any parameterization/mapping. So, the model will choose the lowest $\theta^*$ for convergence. Under these assumptions, then, gradient descent will thus converge to a unique solution $\theta^*$ with corresponding invertible mappings ($f_{\theta^*}, g_{\theta^*}$) between domains $X$ and $Y$.

Given a particular network architecture, then, if the human-preferred canonical map is suboptimal in perception-distortion, the model will converge to a different, possibly unusable solution. To avoid such behavior, small amounts of paired data can produce improved results.

## 7.2 Training Details

We provide a PyTorch implementation of our models.

The dataset consists of satellite photographs and images. We adopt train and test datasets from Zhu et al. (2017a), consisting of 1096 maps and satellite (aerial) photographs. Images are resized with random crop and flip to $256 \times 256$, then normalized before training. For all probabilistic models, we set $\lambda_{cycle} = 10$, $\lambda_{id} = 5$, and $\lambda_{kl} = 1e - 05$ in Equation 9. We use the Adam optimizer (Kingma & Ba, 2015) with a batch size of 5. All networks were trained from scratch with a learning rate of 0.0002 and a latent dimension of 64x16x16 (channels, height, and width respectively) for 600 epochs.

## 7.3 Network Architecture

**Generator:**

We use a U-Net style architecture (Johnson et al., 2016) with a variational bottleneck. The encoder and decoder are symmetric. Let `c7s1−k` denote a $7 \times 7$ Convolution-InstanceNorm-ReLU layer with $k$ filters and stride 1. `d`$k$ denotes a custom Downsampling block via PixelUnshuffle with output channels $k$. `R`$k$ denotes a residual block that contains Reflection padding, two $3 \times 3$ convolutional layers with the same number of filters, two InstanceNorm layers and a ReLU activation.

`L`$k$ denotes a linear ($1 \times 1$ convolutional) layer. `S`$k$ denotes a skip connection block that performs a linear projection or averaging to change the number of channels from the previous layer to $k$. `u`$k$ denotes a custom Upsampling block via Pixelshuffle with output channels $k$. The network uses Reflection Padding throughout to reduce artifacts.

The model encodes an input image into parameters of a Gaussian distribution (mean, $\mu$ and log-variance, $\log \sigma$) in a learned latent space. The dimensionality of this space, $N_z$, determines the size of the bottleneck and the complexity of the learned representation. A latent code $z$ is sampled

from this distribution using the reparameterization trick and is subsequently decoded to generate the output image.

For a $256 \times 256$ input image with 4 downsampling/upsampling layers and one residual block, the architecture is as follows:

**Encoder:** `c7s1-64, d128, d256, d512, d1024, R1024, S`$N_z$

**Variational Bottleneck:**

$\mu = \mathtt{S}_{N_z}(\mathtt{S}_{N_z}(\texttt{enc})),\ \log\sigma^2 = \mathtt{L}_{N_z}(\texttt{enc}),\ \mathbf{z} \sim \mathcal{N}(\mu, \exp(\log\sigma^2))$

**Decoder:** `S1024, R1024, u512, u256, u128, u64, c7s1-3`.

The final output layer (`c7s1-3`) consists of: ReflectionPad2d(3) $\rightarrow$ Conv7-1 $\rightarrow$ Tanh() activation.

**Discriminator:**

The discriminator architecture is adopted from Zhu et al. (2017a), specifically utilizing a 70×70 PatchGAN. The network is constructed from a series of 4×4 convolutional layers with Instance Normalization and LeakyReLU (slope 0.2). The first layer, C64 (64 filters, stride 2), omits Instance Normalization. This is followed by successive layers (C128, C256, C512), each doubling the number of filters. A final convolution layer produces a 1-dimensional output map.

## 7.4 Translation and Reconstruction Examples

Each set of figures/images visualize (a) the aerial to map translation and aerial reconstruction and (b) map to aerial translation and map reconstruction. The maps translated by the Cycle-AE and Cycle-VAE models in an unpaired data setting show no structural similarity to the map domain, as there is neither an adversarial constraint nor a paired dataset to ensure domain alignment. However, spatial information is preserved and allows for the complete reconstruction of the original aerial input. We notice a similar pattern for aerial photos translation and map reconstruction.

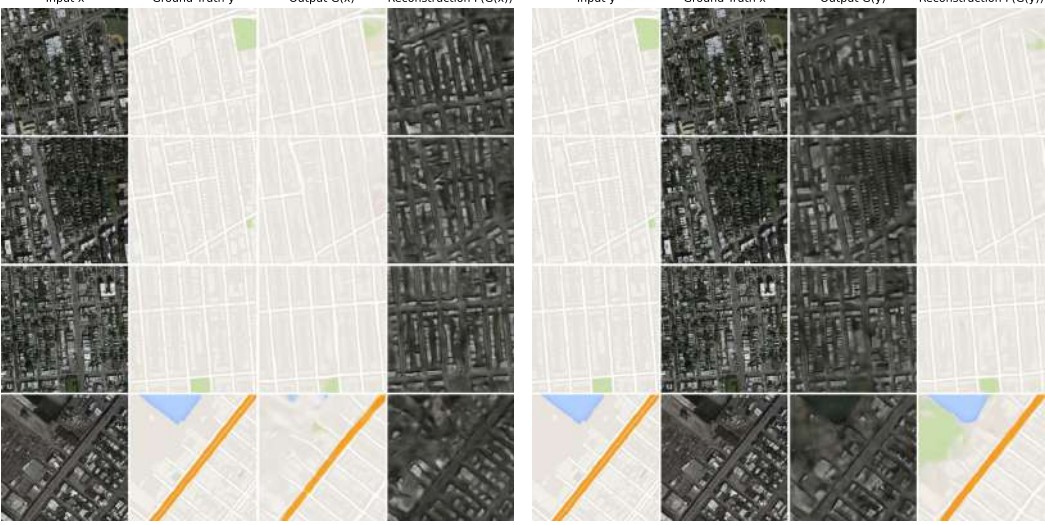

(a) Aerial to map translation and aerial reconstruction    (b) Map to aerial translation and map reconstruction

Figure 12: AE translation and reconstruction.

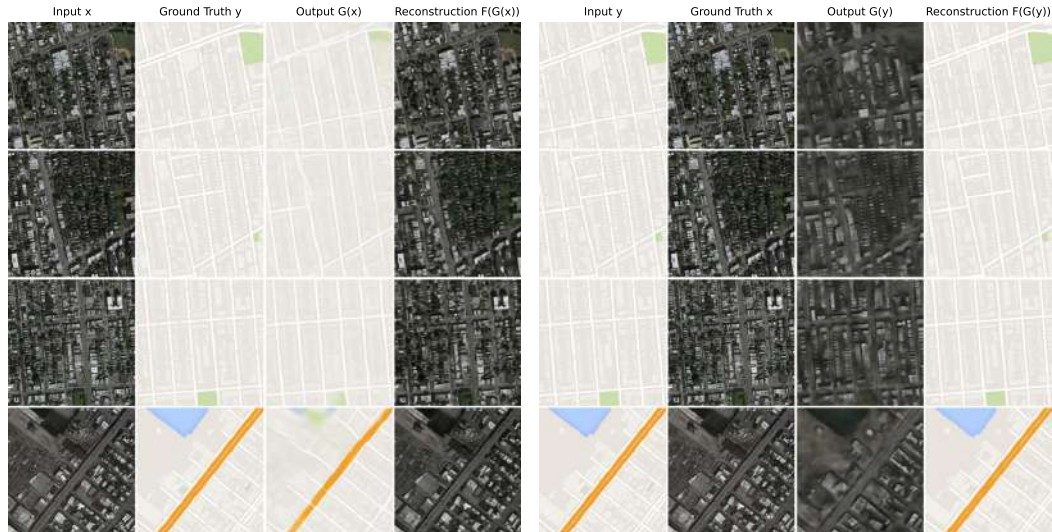

(a) Aerial to map translation and aerial reconstruction    (b) Map to aerial translation and map reconstruction

Figure 13: Cycle-AE (paired) translation and reconstruction.

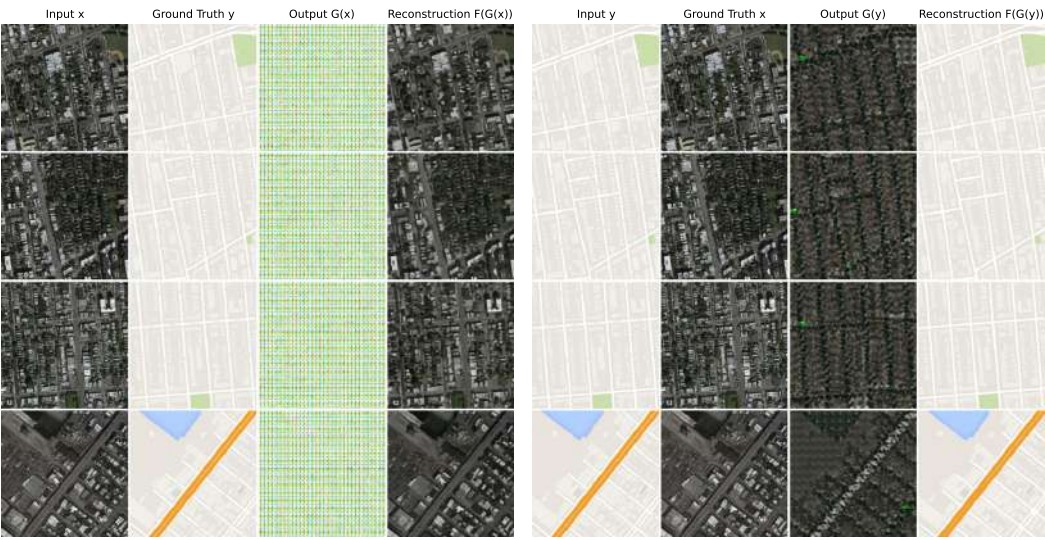

(a) Aerial to map translation and aerial reconstruction    (b) Map to aerial translation and map reconstruction

Figure 14: Cycle-AE (unpaired) translation and reconstruction.

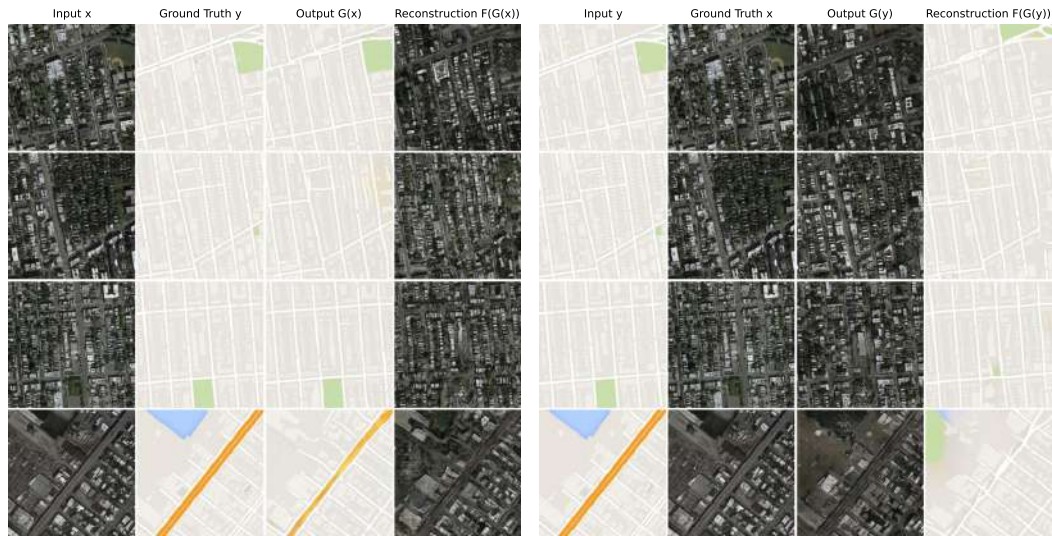

(a) Aerial to map translation and aerial reconstruction    (b) Map to aerial translation and map reconstruction

Figure 15: AE-GAN translation and reconstruction.

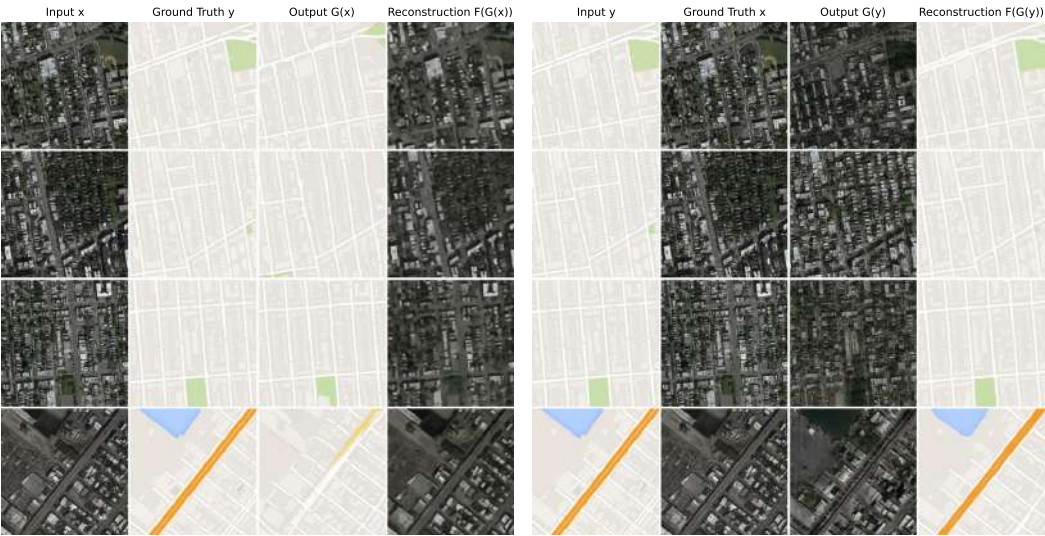

(a) Aerial to map translation and aerial reconstruction    (b) Map to aerial translation and map reconstruction

Figure 16: AE CycleGAN translation and reconstruction.

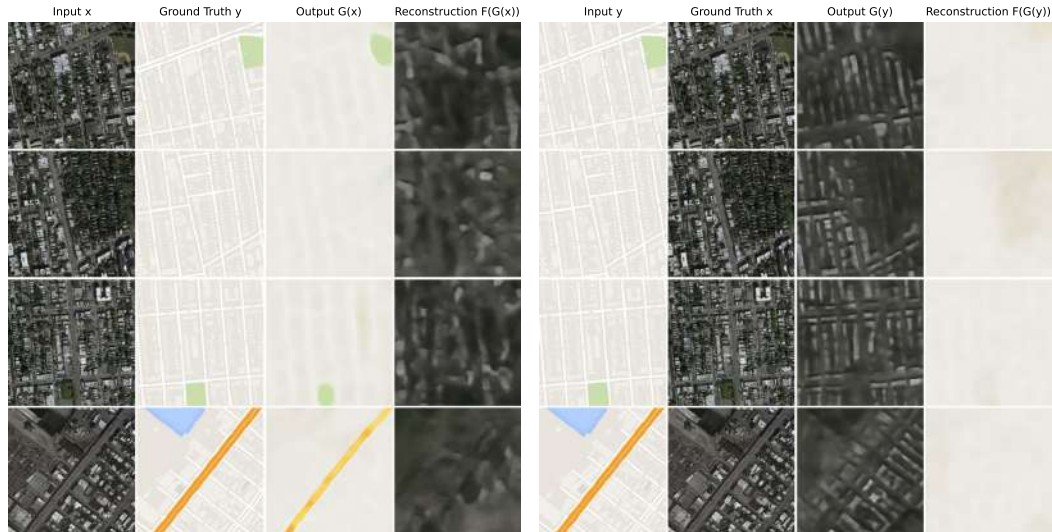

(a) Aerial to map translation and aerial reconstruction    (b) Map to aerial translation and map reconstruction

Figure 17: VAE translation and reconstruction.

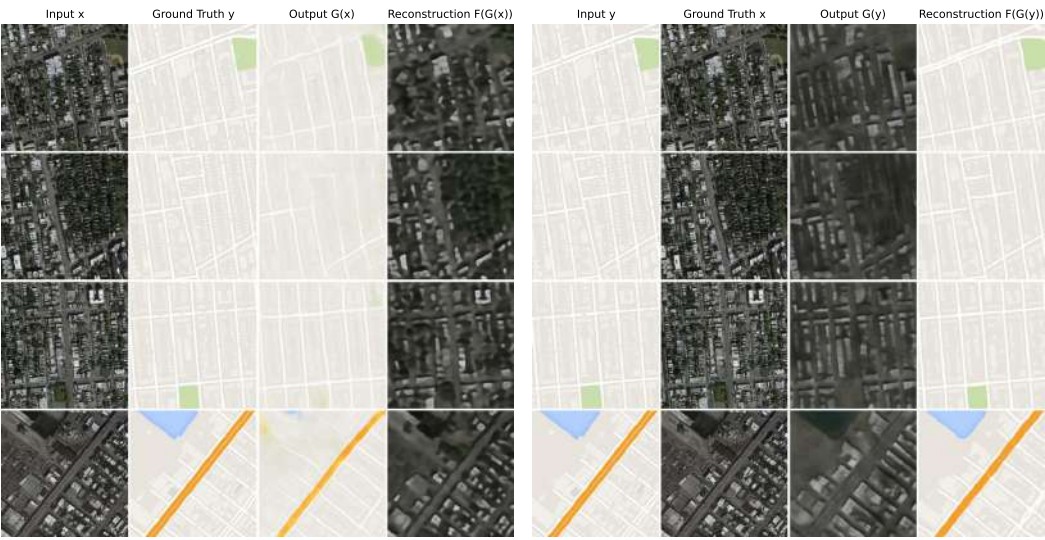

(a) Aerial to map translation and aerial reconstruction    (b) Map to aerial translation and map reconstruction

Figure 18: Cycle-VAE (paired) translation and reconstruction.

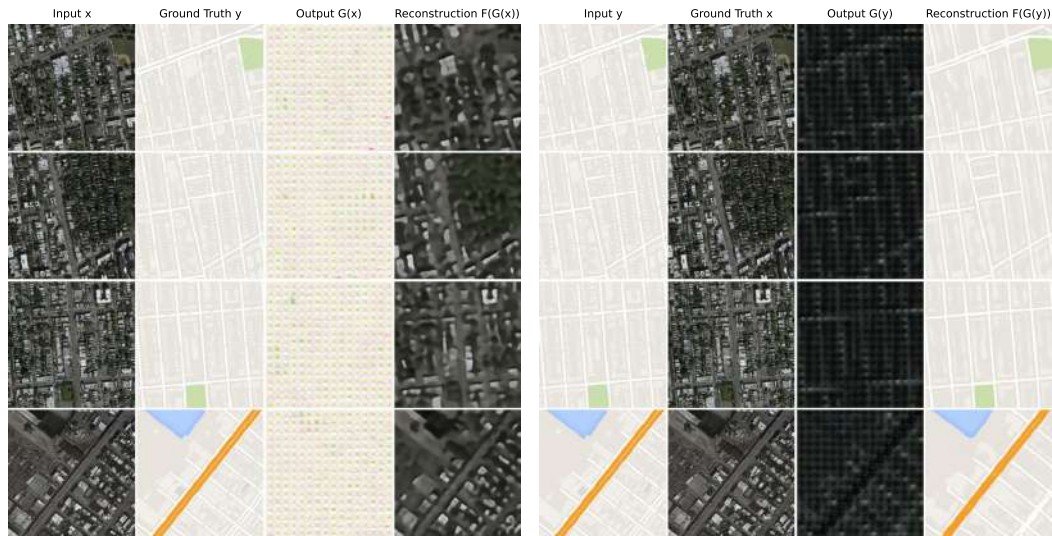

(a) Aerial to map translation and aerial reconstruction    (b) Map to aerial translation and map reconstruction

Figure 19: Cycle-VAE (unpaired) translation and reconstruction.

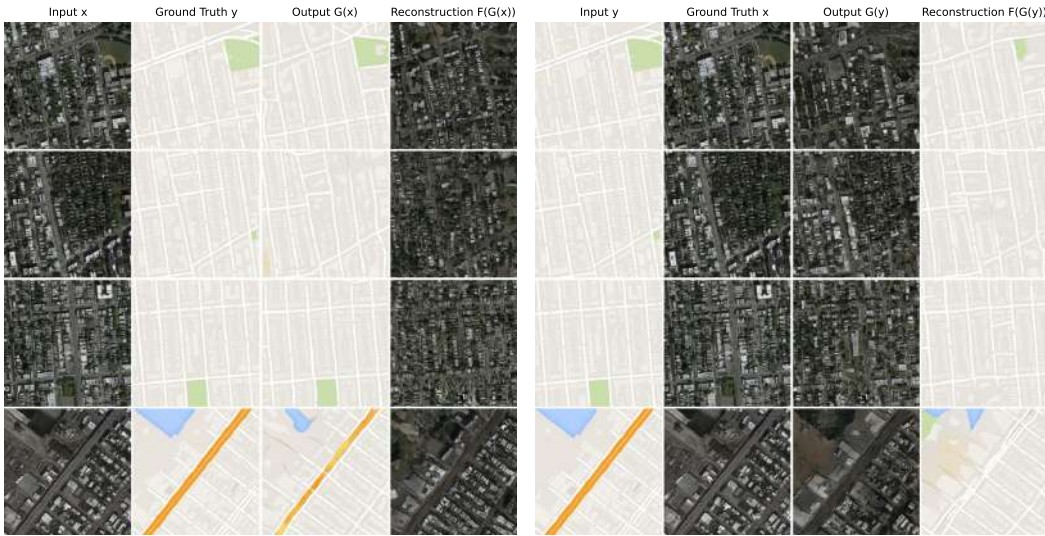

(a) Aerial to map translation and aerial reconstruction    (b) Map to aerial translation and map reconstruction

Figure 20: VAE-GAN translation and reconstruction.

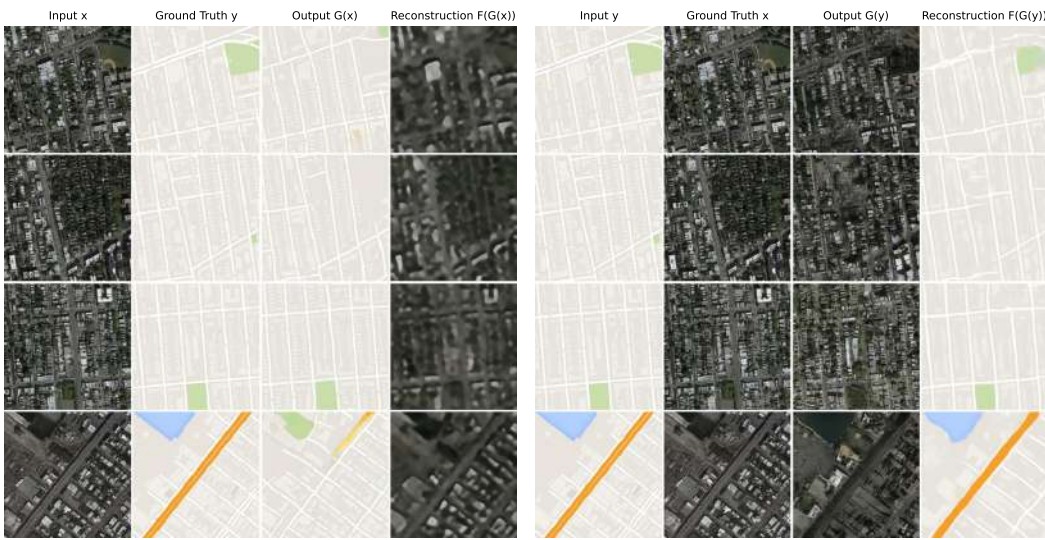

(a) Aerial to map translation and aerial reconstruction   (b) Map to aerial translation and map reconstruction

Figure 21: VAE CycleGAN translation and reconstruction.

### 7.4.1 ERROR EVALUATION METRICS

Table 5: Translation and reconstruction error evaluation of Autoencoder (AE) based models on aerial photos ($x$) $\leftrightarrow$ maps ($y$) after 600 epochs

(a) AE

| AE | MSE | PSNR | SSIM | FID | KID Mean | KID STD |
|---|---|---|---|---|---|---|
| Aerial Translation | 0.0256 | 14.08 | 0.2489 | 253.05 | **0.2331** | **0.0196** |
| Map Translation | **0.0018** | **27.37** | **0.7636** | 325.56 | 0.3530 | 0.0374 |
| Aerial Reconstruction | 0.0287 | 13.95 | 0.1992 | 288.35 | **0.2807** | **0.0208** |
| Map Reconstruction | **0.0017** | **27.77** | **0.7718** | 315.15 | 0.3347 | 0.0324 |

(b) CycleAE-paired

| CycleAE-paired | MSE | PSNR | SSIM | FID | KID Mean | KID STD |
|---|---|---|---|---|---|---|
| Aerial Translation | 0.0251 | 14.13 | 0.2553 | 241.77 | **0.2175** | **0.0189** |
| Map Translation | **0.0018** | **27.46** | **0.7650** | 273.17 | 0.2729 | 0.0241 |
| Aerial Reconstruction | 0.0059 | 21.53 | 0.6661 | 175.29 | 0.1621 | 0.0164 |
| Map Reconstruction | **0.0003** | **35.91** | **0.9242** | **83.08** | **0.0604** | **0.0066** |

(c) CycleAE-unpaired

| CycleAE-unpaired | MSE | PSNR | SSIM | FID | KID Mean | KID STD |
|---|---|---|---|---|---|---|
| Aerial Translation | **0.0410** | **13.69** | **0.0924** | 329.51 | **0.3552** | 0.0175 |
| Map Translation | 0.0872 | 10.59 | 0.0318 | 409.27 | 0.5077 | **0.0163** |
| Aerial Reconstruction | 0.0063 | 21.12 | 0.6451 | 180.12 | 0.1662 | 0.0188 |
| Map Reconstruction | **0.0002** | **36.59** | **0.9317** | **80.05** | **0.0608** | **0.0075** |

(d) AE-GAN

| AE-GAN | MSE | PSNR | SSIM | FID | KID Mean | KID STD |
|---|---|---|---|---|---|---|
| Aerial Translation | 0.0349 | 14.29 | 0.2079 | **52.33** | **0.0175** | **0.0055** |
| Map Translation | **0.0031** | **25.10** | **0.6684** | 63.83 | 0.0316 | 0.0066 |
| Aerial Reconstruction | 0.0392 | 13.90 | 0.1481 | **70.75** | **0.0347** | **0.0067** |
| Map Reconstruction | **0.0050** | **22.99** | **0.6969** | 74.76 | 0.0425 | 0.0082 |

(e) AE-CycleGAN

| AE-CycleGAN | MSE | PSNR | SSIM | FID | KID Mean | KID STD |
|---|---|---|---|---|---|---|
| Aerial Translation | 0.0389 | 13.62 | 0.1641 | **52.53** | **0.0170** | **0.0048** |
| Map Translation | **0.0050** | **22.97** | **0.6737** | 70.08 | 0.0460 | 0.0131 |
| Aerial Reconstruction | 0.0096 | 19.01 | 0.5069 | 200.11 | 0.1833 | 0.0181 |
| Map Reconstruction | **0.0005** | **32.99** | **0.8760** | 106.63 | **0.0844** | **0.0092** |

Table 6: Translation and reconstruction error evaluation of Variational Autoencoder (VAE) based models on aerial photos ($x$) $\leftrightarrow$ maps ($y$) after 600 epochs.

(a) VAE

| VAE | MSE | PSNR | SSIM | FID | KID Mean | KID STD |
|---|---|---|---|---|---|---|
| Aerial Translation | 0.0261 | 13.50 | 0.1894 | 304.34 | **0.2858** | **0.0209** |
| Map Translation | **0.0024** | **26.03** | **0.6753** | 358.70 | 0.3547 | 0.0310 |
| Aerial Reconstruction | 0.0340 | 11.13 | 0.1344 | 347.17 | **0.3642** | **0.0188** |
| Map Reconstruction | **0.0049** | **22.63** | **0.6717** | 367.13 | 0.3895 | 0.0250 |

(b) Cycle-VAE-paired

| Cycle-VAE-paired | MSE | PSNR | SSIM | FID | KID Mean | KID STD |
|---|---|---|---|---|---|---|
| Aerial Translation | 0.0255 | 13.25 | 0.2080 | 259.83 | **0.2403** | **0.0189** |
| Map Translation | **0.0022** | **26.58** | **0.7117** | 325.89 | 0.3369 | 0.0236 |
| Aerial Reconstruction | 0.0145 | 17.03 | 0.3330 | 266.76 | 0.2587 | 0.0173 |
| Map Reconstruction | **0.0007** | **31.81** | **0.8450** | 170.27 | **0.1546** | **0.0154** |

(c) Cycle-VAE-unpaired

| Cycle-VAE-unpaired | MSE | PSNR | SSIM | FID | KID Mean | KID STD |
|---|---|---|---|---|---|---|
| Aerial Translation | 0.0614 | 7.90 | 0.0885 | 475.98 | 0.5931 | 0.0274 |
| Map Translation | **0.0113** | **19.47** | **0.3128** | 419.51 | **0.5393** | **0.0194** |
| Aerial Reconstruction | 0.0146 | 17.31 | 0.3151 | 281.64 | 0.2761 | **0.0168** |
| Map Reconstruction | **0.0006** | **32.44** | **0.8468** | 219.37 | **0.2130** | 0.0282 |

(d) VAE-GAN

| VAE-GAN | MSE | PSNR | SSIM | FID | KID Mean | KID STD |
|---|---|---|---|---|---|---|
| Aerial Translation | 0.0354 | 14.17 | 0.1705 | **64.45** | **0.0301** | **0.0052** |
| Map Translation | **0.0033** | **24.85** | **0.6272** | 83.18 | 0.0510 | 0.0091 |
| Aerial Reconstruction | 0.0430 | 13.35 | 0.0954 | **84.63** | **0.0496** | **0.0101** |
| Map Reconstruction | **0.0060** | **22.22** | **0.6172** | 104.49 | 0.0800 | 0.0107 |

(e) VAE-CycleGAN

| VAE-CycleGAN | MSE | PSNR | SSIM | FID | KID Mean | KID STD |
|---|---|---|---|---|---|---|
| Aerial Translation | 0.0443 | 13.36 | 0.0965 | **69.25** | **0.0378** | **0.0063** |
| Map Translation | **0.0056** | **22.50** | **0.5793** | 90.87 | 0.0594 | 0.0092 |
| Aerial Reconstruction | 0.0175 | 16.10 | 0.2779 | 271.72 | 0.2703 | **0.0200** |
| Map Reconstruction | **0.0011** | **29.67** | **0.7804** | 241.78 | **0.2409** | 0.0259 |

### 7.4.2 COMPARISON: AE VARIANTS

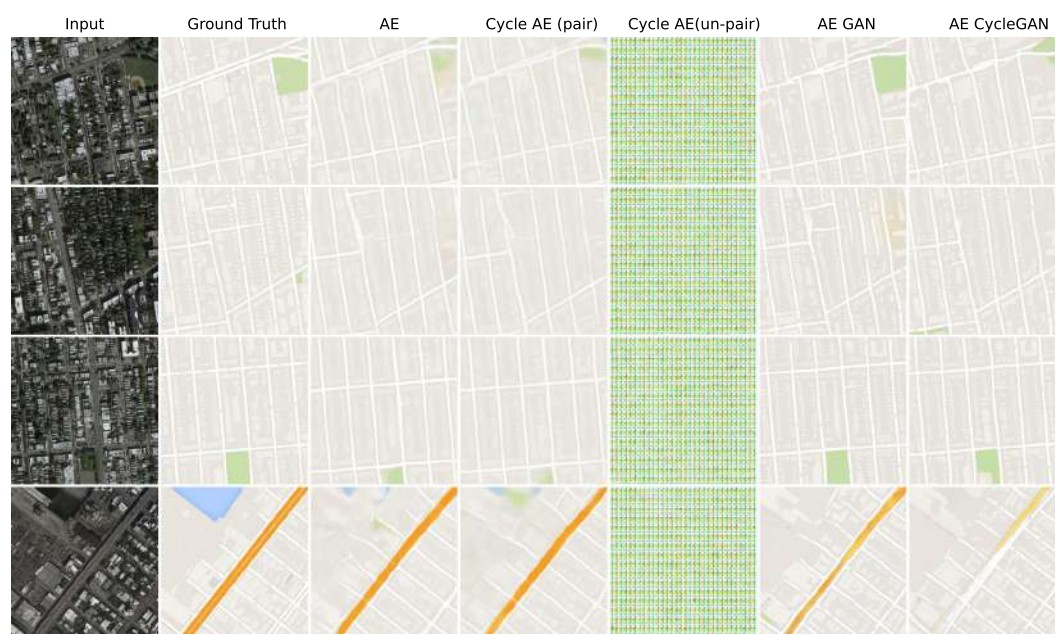

Figure 22: Translated maps $x \to G(x)$ of different AE models for the given aerial input $x$. From left to right: input, ground truth, translated map output from models AE, Cycle-AE (paired data), Cycle-AE (unpaired data), AE-GAN, and AE-CycleGAN.

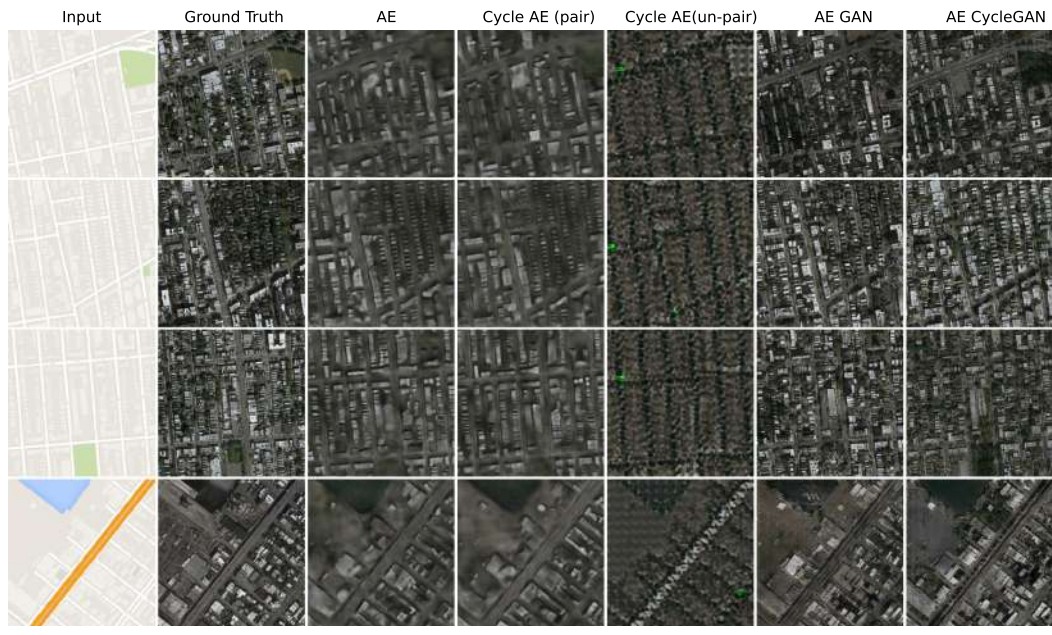

Figure 23: Translated aerial images $y \to F(y)$ of different AE models for the given map input $y$. . From left to right: input, ground truth, translated aerial images from models AE, Cycle-AE (paired data), Cycle-AE (unpaired data), AE-GAN, and AE-CycleGAN.

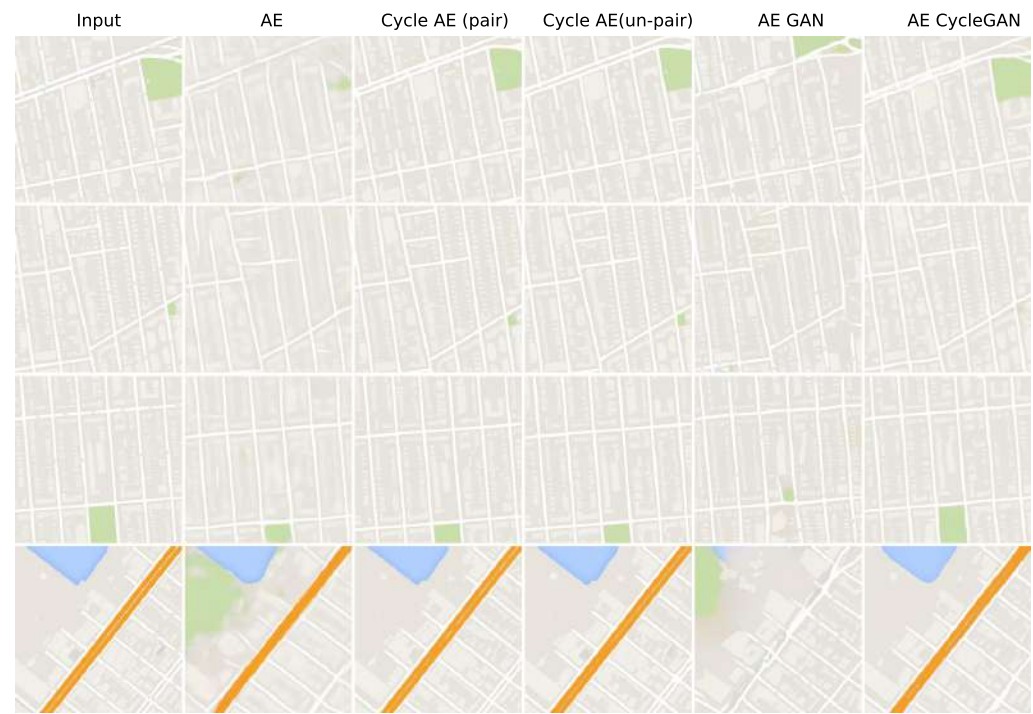

Figure 24: Reconstructed maps $G(F(y))$ of different AE models for the given input $y$. From left to right: input $y$, reconstructed maps from the models AE, Cycle-AE (paired data), Cycle-AE (unpaired data), AE-GAN, and AE-CycleGAN.

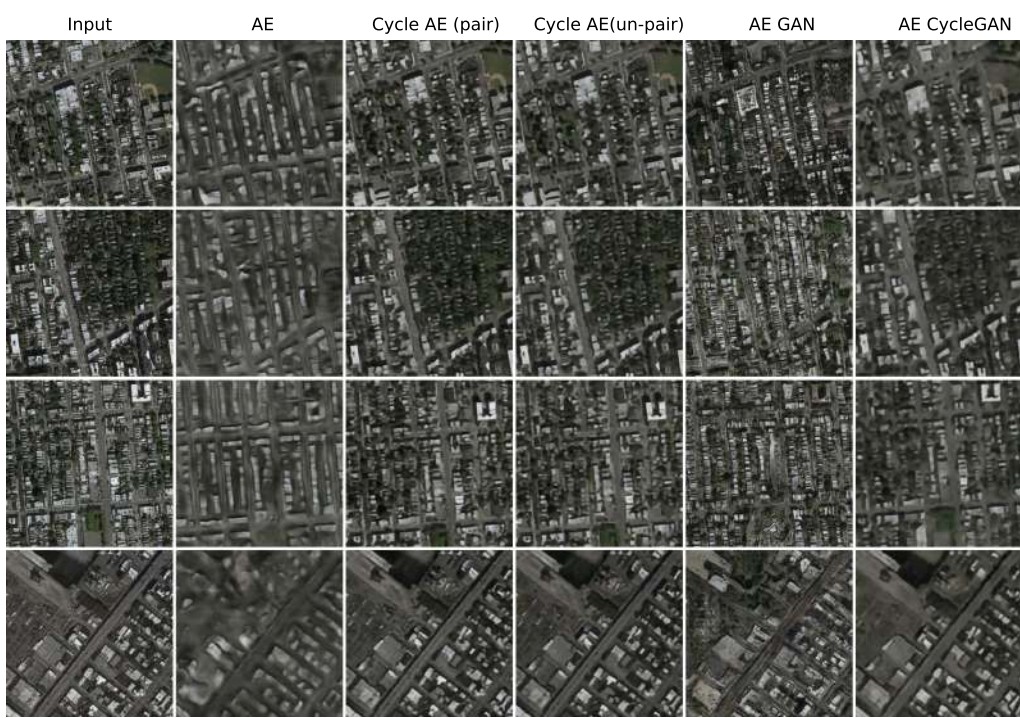

Figure 25: Reconstructed aerial images $F(G(x))$ of different AE models for the given input $x$. From left to right: input $x$, reconstructed maps from the models AE, Cycle-AE (paired data), Cycle-AE (unpaired data), AE-GAN, and AE-CycleGAN.

### 7.4.3 COMPARISON: VAE VARIANTS

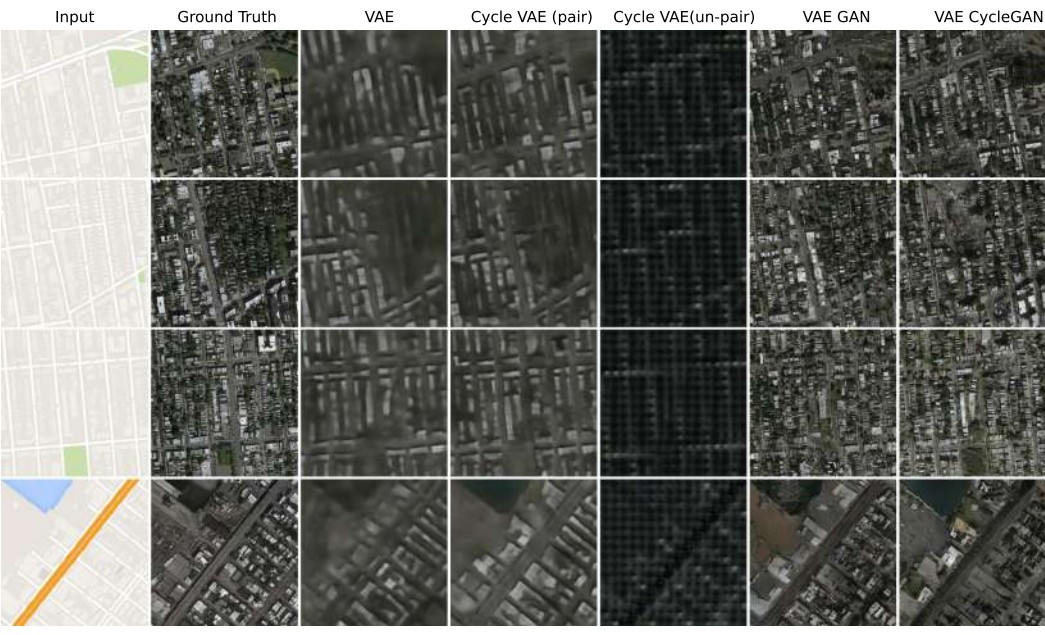

Figure 26: Translated maps $x \to G(x)$ of different VAE models for the given aerial input $x$. From left to right: input, ground truth, translated map output from models VAE, Cycle-VAE (paired data), Cycle-VAE (unpaired data), VAE-GAN, and VAE-CycleGAN.

Figure 27: Translated aerial images $y \to F(y)$ of different VAE models for the given map input $y$. From left to right: input, ground truth, translated aerial images from models VAE, Cycle-VAE (paired data), Cycle-VAE (unpaired data), VAE-GAN, and VAE-CycleGAN.

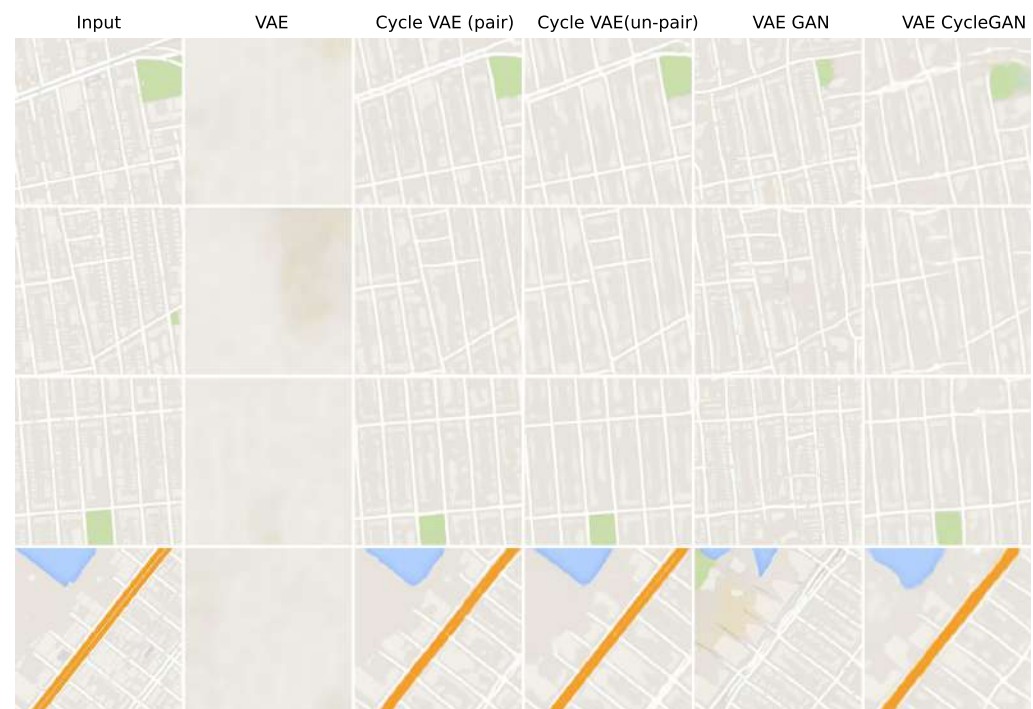

Figure 28: Reconstructed maps $G(F(y))$ of different VAE models for the given input $y$. From left to right: input $y$, reconstructed maps from the models VAE, Cycle-VAE (paired data), Cycle-VAE (unpaired data), VAE-GAN, and VAE-CycleGAN.

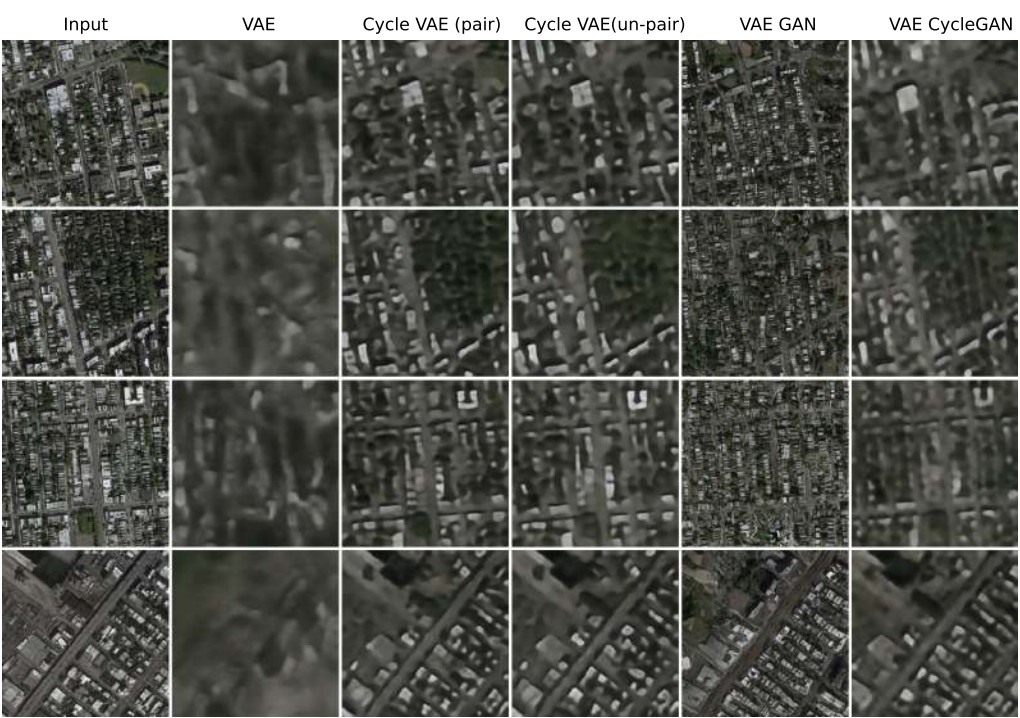

Figure 29: Reconstructed aerial images $F(G(x))$ of different VAE models for the given input $x$. From left to right: input $x$, reconstructed aerial images from the models VAE, Cycle-VAE (paired data), Cycle-VAE (unpaired data), VAE-GAN, and VAE-CycleGAN.

### 7.4.4 COMPARISON BETWEEN AE AND VAE MODELS

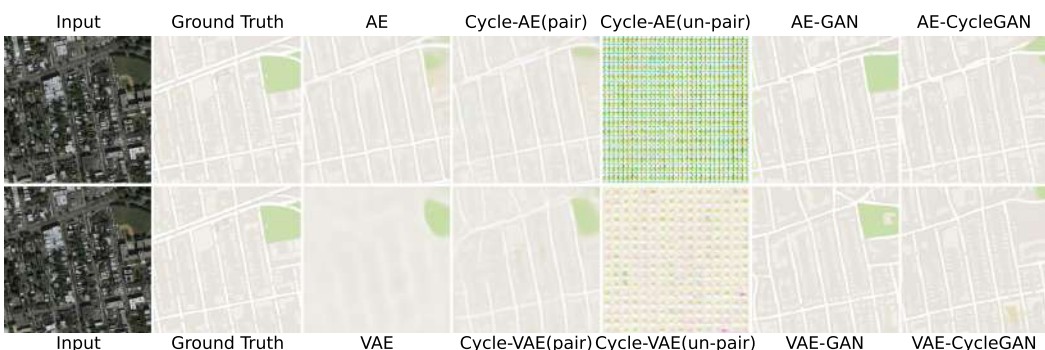

Figure 30: Comparison of translated map outputs $G(x)$ from different models for a given input $x$. From left to right: input $x$, ground truth $y$, translated map output from the models. From top to bottom: AE variants, VAE variants.

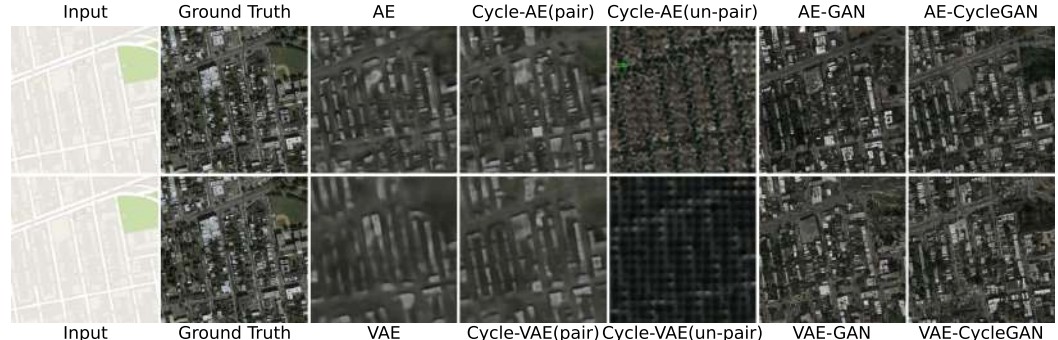

Figure 31: Comparison of translated aerial image outputs $F(y)$ from different models for a given input $y$. From left to right: input $y$, ground truth $x$, translated aerial output from the models. From top to bottom: AE variants, VAE variants.

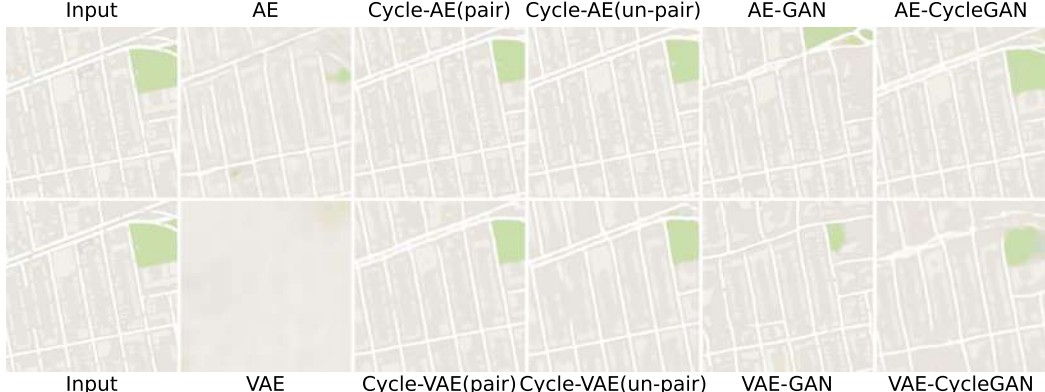

Figure 32: Comparison of reconstructed maps $G(F(y))$ from different models for a given input $y$. From left to right: input $y$, reconstructed map output from the models. From top to bottom: AE variants, VAE variants.

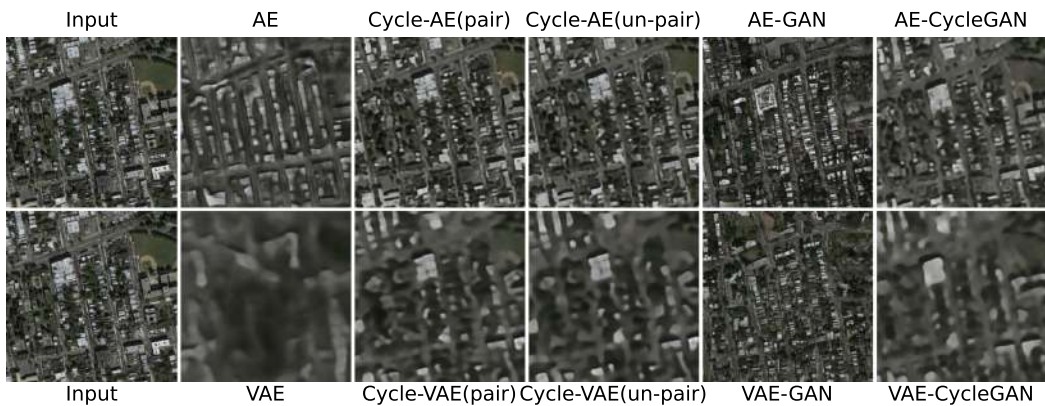

Figure 33: Comparison of reconstructed aerial image outputs $F(G(x))$ from different models for a given input $y$. From left to right: input $x$, reconstructed aerial output from the models. From top to bottom: AE variants, VAE variants.

### 7.4.5 ERROR EVALUATION: TRANSLATION AND RECONSTRUCTION

Table 7: Translation error evaluation: comparison between deterministic and stochastic models.

| | Model | $G : X \rightarrow Y$ (aerial→map) | | | | | $F : Y \rightarrow X$ (map→aerial) | | | | |
|---|---|---|---|---|---|---|---|---|---|---|---|
| | | MSE ↓ | PSNR ↑ | SSIM ↑ | FID ↓ | KID $\mu \pm \sigma$ ↓ | MSE ↓ | PSNR ↑ | SSIM ↑ | FID ↓ | KID $\mu \pm \sigma$ ↓ |
| **Deterministic** | AE | **0.0018** | 27.37 | 0.7636 | 325.56 | $0.3530 \pm 0.0374$ | 0.0256 | 14.08 | 0.2489 | 253.05 | $0.2331 \pm 0.0196$ |
| | Cycle AE (paired) | **0.0018** | **27.46** | **0.7650** | 273.17 | $0.2729 \pm 0.0241$ | **0.0251** | 14.13 | **0.2553** | 241.77 | $0.2175 \pm 0.0189$ |
| | Cycle AE (unpaired) | 0.0872 | 10.59 | 0.0318 | 409.27 | $0.5077 \pm 0.0163$ | 0.0410 | 13.69 | 0.0924 | 329.51 | $0.3552 \pm 0.0175$ |
| | AE-GAN | 0.0031 | 25.10 | 0.6684 | **63.83** | $\mathbf{0.0316 \pm 0.0066}$ | 0.0349 | **14.29** | 0.2079 | **52.33** | $0.0175 \pm 0.0055$ |
| | AE-CycleGAN | 0.0050 | 22.97 | 0.6737 | 70.08 | $0.0460 \pm 0.0131$ | 0.0389 | 13.62 | 0.1641 | 52.53 | $\mathbf{0.0170 \pm 0.0048}$ |
| **Stochastic** | VAE | 0.0024 | 26.03 | 0.6753 | 358.70 | $0.3547 \pm 0.0310$ | 0.0261 | 13.50 | 0.1894 | 304.34 | $0.2858 \pm 0.0209$ |
| | Cycle VAE (paired) | **0.0022** | **26.58** | **0.7117** | 325.89 | $0.3369 \pm 0.0236$ | **0.0255** | 13.25 | **0.2080** | 259.83 | $0.2403 \pm 0.0189$ |
| | Cycle VAE (unpaired) | 0.0113 | 19.47 | 0.3128 | 419.51 | $0.5393 \pm 0.0194$ | 0.0614 | 7.90 | 0.0885 | 475.98 | $0.5931 \pm 0.0274$ |
| | VAE-GAN | 0.0033 | 24.85 | 0.6272 | **83.18** | $\mathbf{0.0510 \pm 0.0091}$ | 0.0354 | **14.17** | 0.1705 | **64.45** | $\mathbf{0.0301 \pm 0.0052}$ |
| | VAE-CycleGAN | 0.0056 | 22.50 | 0.5793 | 90.87 | $0.0594 \pm 0.0092$ | 0.0443 | 13.36 | 0.0965 | 69.25 | $0.0378 \pm 0.0063$ |

Table 8: Reconstruction error: comparison between deterministic and stochastic models.

| | Model | $G(F(y))$: Map Reconstruction | | | | | $F(G(x))$: Aerial Reconstruction | | | | |
|---|---|---|---|---|---|---|---|---|---|---|---|
| | | MSE ↓ | PSNR ↑ | SSIM ↑ | FID ↓ | KID $\mu \pm \sigma$ ↓ | MSE ↓ | PSNR ↑ | SSIM ↑ | FID ↓ | KID $\mu \pm \sigma$ ↓ |
| **Deterministic** | AE | 0.0017 | 27.77 | 0.7718 | 315.15 | $0.3347 \pm 0.0324$ | 0.0287 | 13.95 | 0.1992 | 288.35 | $0.2807 \pm 0.0208$ |
| | Cycle AE (paired) | 0.0003 | 35.91 | 0.9242 | 83.08 | $0.0604 \pm 0.0066$ | **0.0059** | **21.53** | **0.6661** | 175.29 | $0.1621 \pm 0.0164$ |
| | Cycle AE (unpaired) | **0.0002** | **36.59** | **0.9317** | 80.05 | $0.0608 \pm 0.0075$ | 0.0063 | 21.12 | 0.6451 | 180.12 | $0.1662 \pm 0.0188$ |
| | AE-GAN | 0.0050 | 22.99 | 0.6969 | **74.76** | $\mathbf{0.0425 \pm 0.0082}$ | 0.0392 | 13.90 | 0.1481 | **70.75** | $\mathbf{0.0347 \pm 0.0067}$ |
| | AE-CycleGAN | 0.0005 | 32.99 | 0.8760 | 106.63 | $0.0844 \pm 0.0092$ | 0.0096 | 19.01 | 0.5069 | 200.11 | $0.1833 \pm 0.0181$ |
| **Stochastic** | VAE | 0.0049 | 22.63 | 0.6717 | 367.13 | $0.3895 \pm 0.0250$ | 0.0340 | 11.13 | 0.1344 | 347.17 | $0.3642 \pm 0.0188$ |
| | Cycle VAE (paired) | 0.0007 | 31.81 | 0.8450 | 170.27 | $0.1546 \pm 0.0154$ | **0.0145** | 17.03 | **0.3330** | 266.76 | $0.2587 \pm 0.0173$ |
| | Cycle VAE (unpaired) | **0.0006** | **32.44** | **0.8468** | 219.37 | $0.2130 \pm 0.0282$ | 0.0146 | **17.31** | 0.3151 | 281.64 | $0.2761 \pm 0.0168$ |
| | VAE-GAN | 0.0060 | 22.22 | 0.6172 | 104.49 | $\mathbf{0.0800 \pm 0.0107}$ | 0.0430 | 13.35 | 0.0954 | **84.63** | $\mathbf{0.0496 \pm 0.0101}$ |
| | VAE-CycleGAN | 0.0011 | 29.67 | 0.7804 | 241.78 | $0.2409 \pm 0.0259$ | 0.0175 | 16.10 | 0.2779 | 271.72 | $0.2703 \pm 0.0200$ |

## 7.5 REALIZATIONS

### 7.5.1 VARIATIONAL AUTOENCODER (VAE) REALIZATIONS

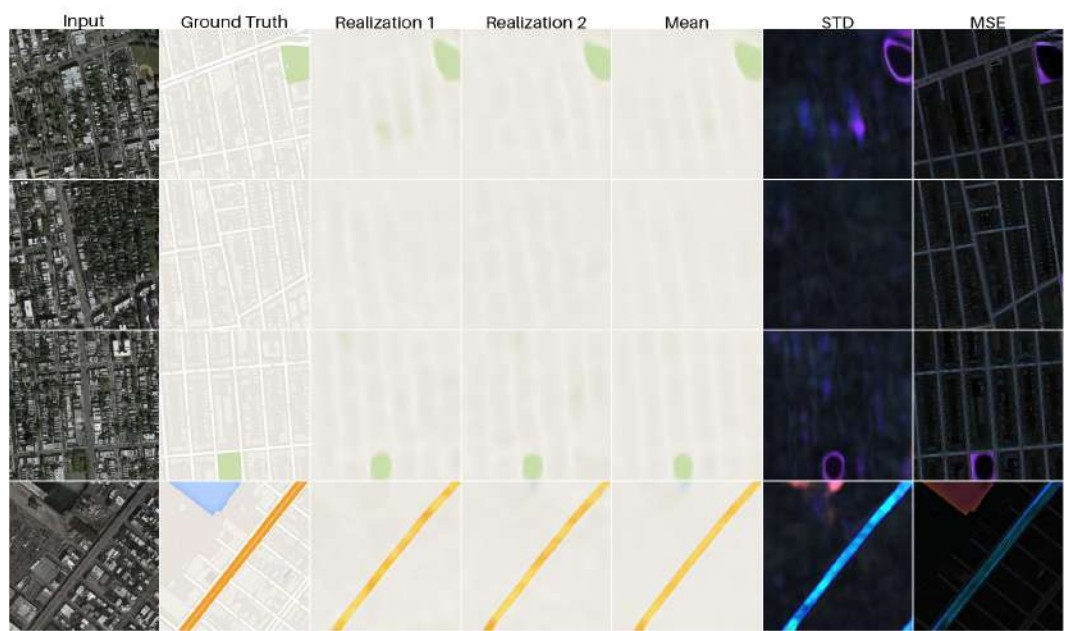

Figure 34: VAE model map realizations. From left to right: input $(x)$, ground truth $(y)$, 2 sample realizations, mean, STD, MSE of the 30 map realizations. STD is scaled by $20\times$ for visibility.

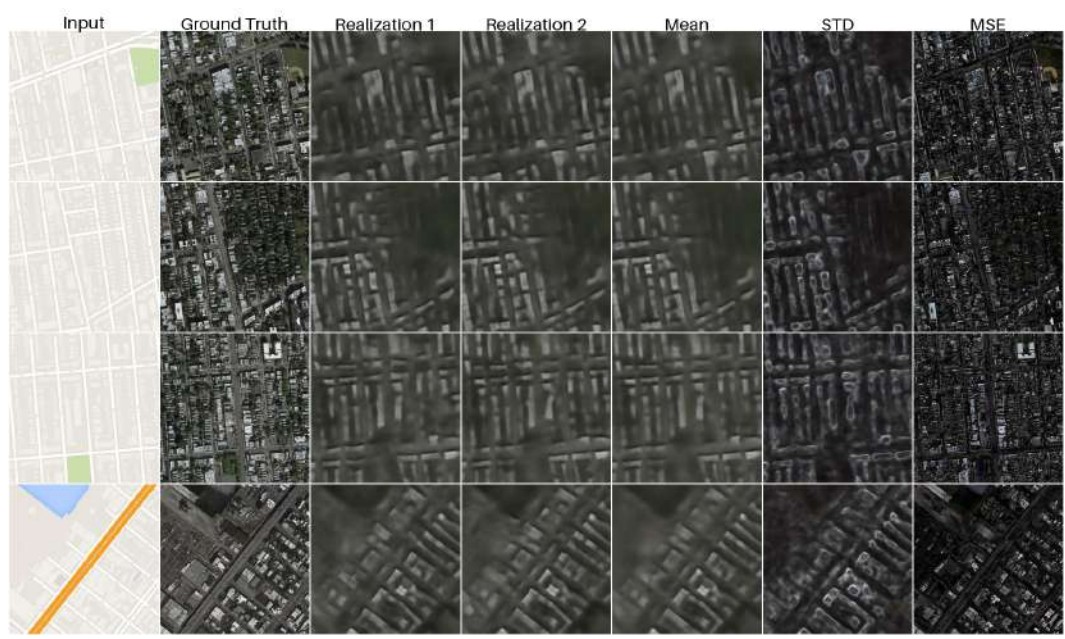

Figure 35: VAE model aerial photo realizations. From left to right: input $(y)$, ground truth $(x)$, 2 sample realizations, mean, STD and MSE of the 30 aerial realizations. STD is scaled by $10\times$ for visibility.

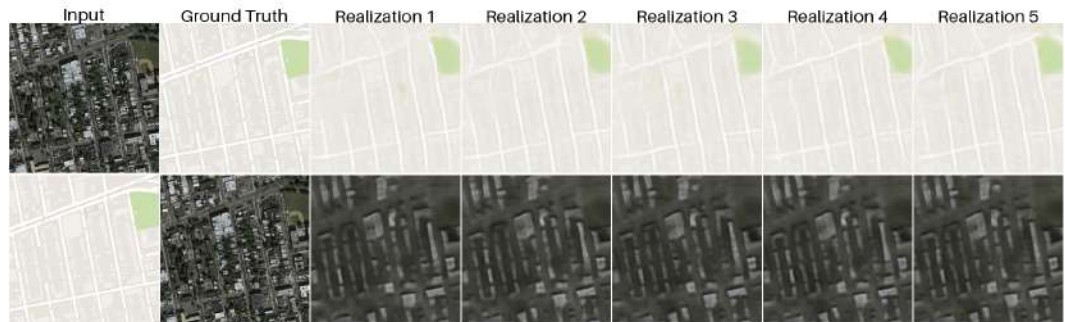

Figure 36: Cycle VAE (paired) sample aerial and map realizations.

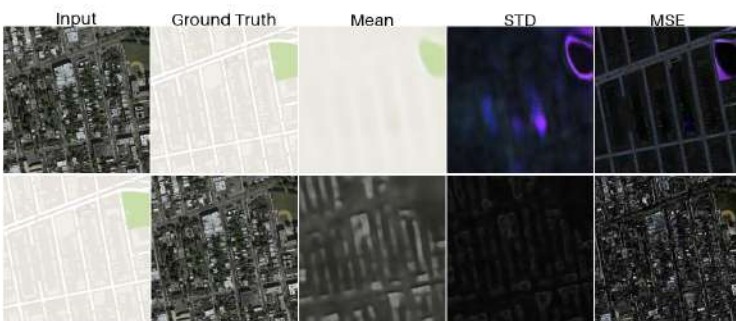

Figure 37: Cycle-VAE (paired) model mean, STD and MSE of the 30 realizations. Top row: aerial→ maps, bottom row: maps → aerial images. For visibility, the STD of the aerial image is scaled by 10x and the STD of the map image by $20\times$.

### 7.5.2 CYCLE-VAE (PAIRED) REALIZATIONS

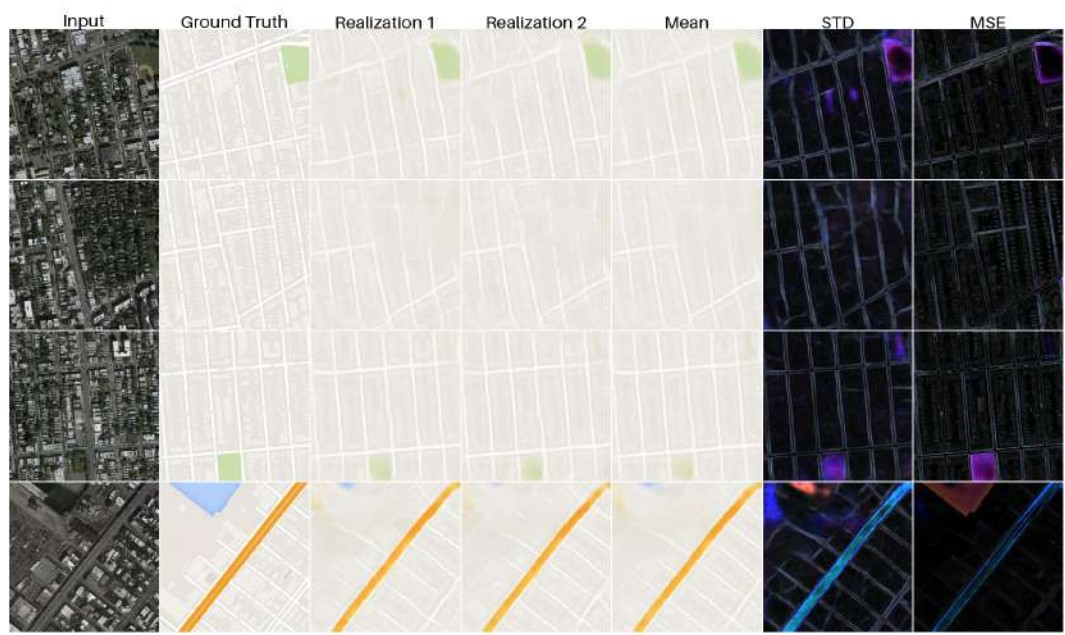

Figure 38: Cycle-VAE (paired) model map realizations. From left to right: input $(x)$, ground truth $(y)$, 2 sample realizations, mean, STD, MSE of the 30 map realizations. STD is scaled by $20\times$ for visibility.

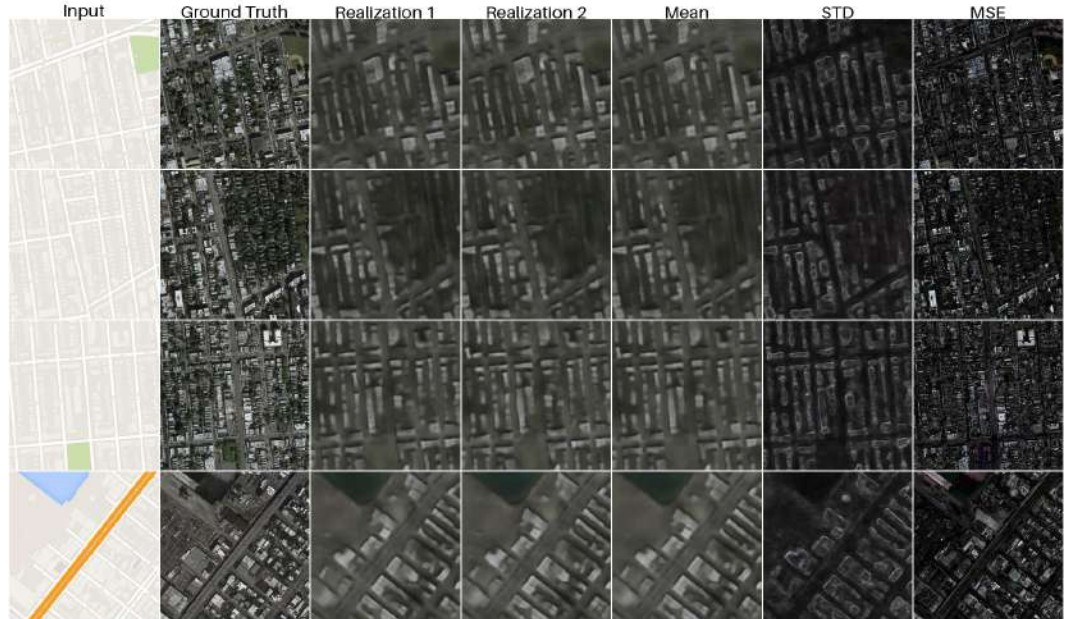

Figure 39: Cycle-VAE (paired) model aerial photo realizations. From left to right: input ($\mathbf{y}$), ground truth ($\mathbf{x}$), 2 sample realizations, mean, STD and MSE of the 30 aerial realizations. STD is scaled by $10\times$ for visibility.

### 7.5.3 CYCLE-VAE (UNPAIRED) REALIZATIONS

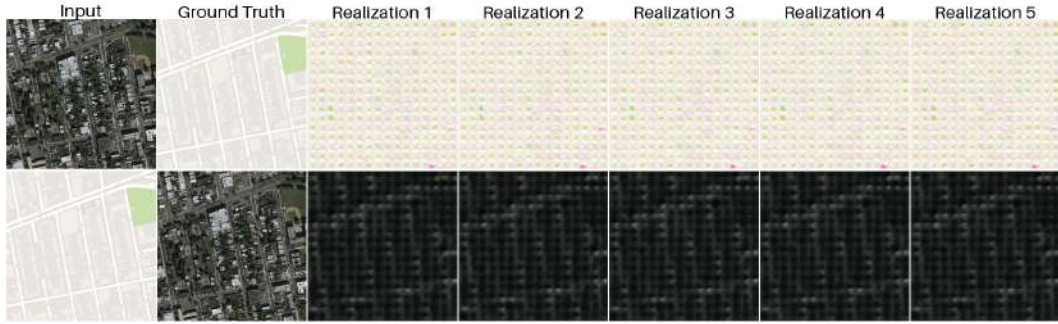

Figure 40: Cycle VAE (unpaired) sample aerial and map realizations.

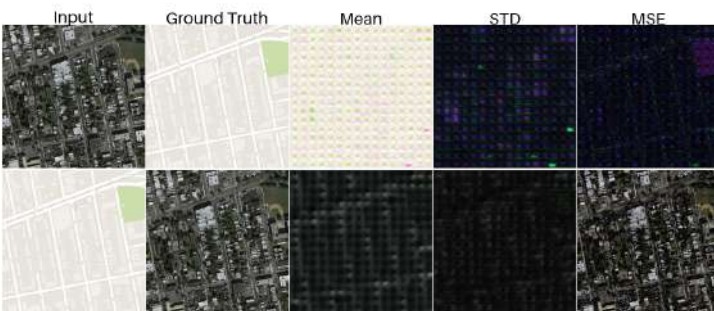

Figure 41: Cycle-VAE (unpaired) model mean, STD and MSE of the 30 realizations. Top row: aerial$\rightarrow$ maps, bottom row: maps $\rightarrow$ aerial images. For visibility, the STD of the aerial image is scaled by $10\times$ and the STD of the map image by $20\times$.

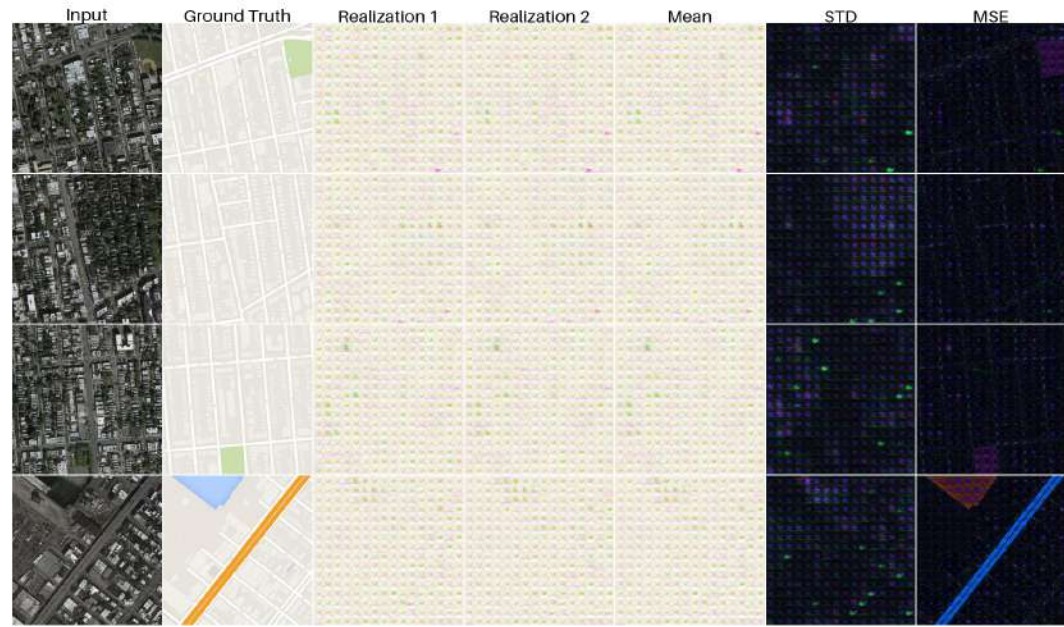

Figure 42: Cycle-VAE (unpaired) model map realizations. From left to right: input $(x)$, ground truth $(y)$, 2 sample realizations, mean, STD, MSE of the 30 map realizations. STD is scaled by $20\times$ for visibility.

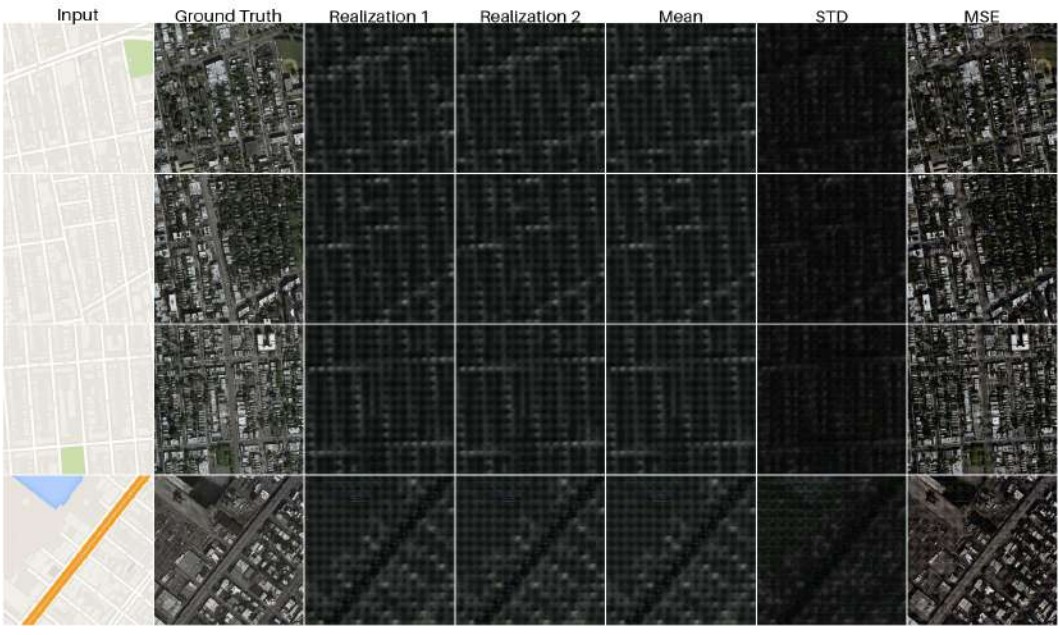

Figure 43: Cycle-VAE (unpaired) model aerial photo realizations. From left to right: input $(y)$, ground truth $(x)$, 2 sample realizations, mean, STD, and MSE of the 30 aerial realizations. STD is scaled by $10\times$ for visibility.

### 7.5.4 VAE-GAN REALIZATIONS

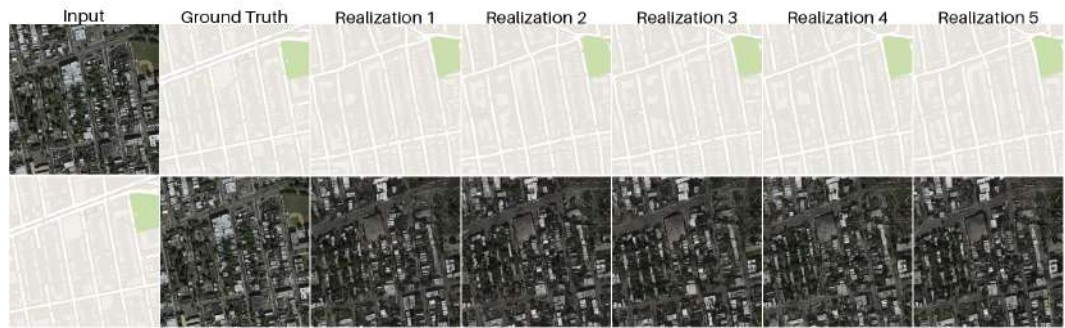

Figure 44: VAE-GAN sample aerial and map realizations.

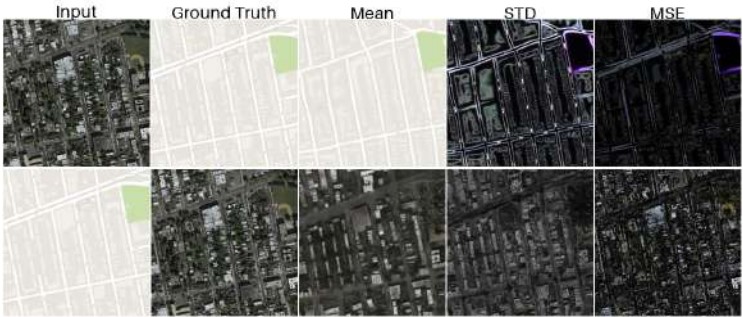

Figure 45: VAE-GAN model mean, STD and MSE of the 30 realizations. Top row: aerial→ maps, bottom row: maps → aerial images. For visibility, the STD of the aerial image is scaled by $3\times$ and the STD of the map image by $20\times$.

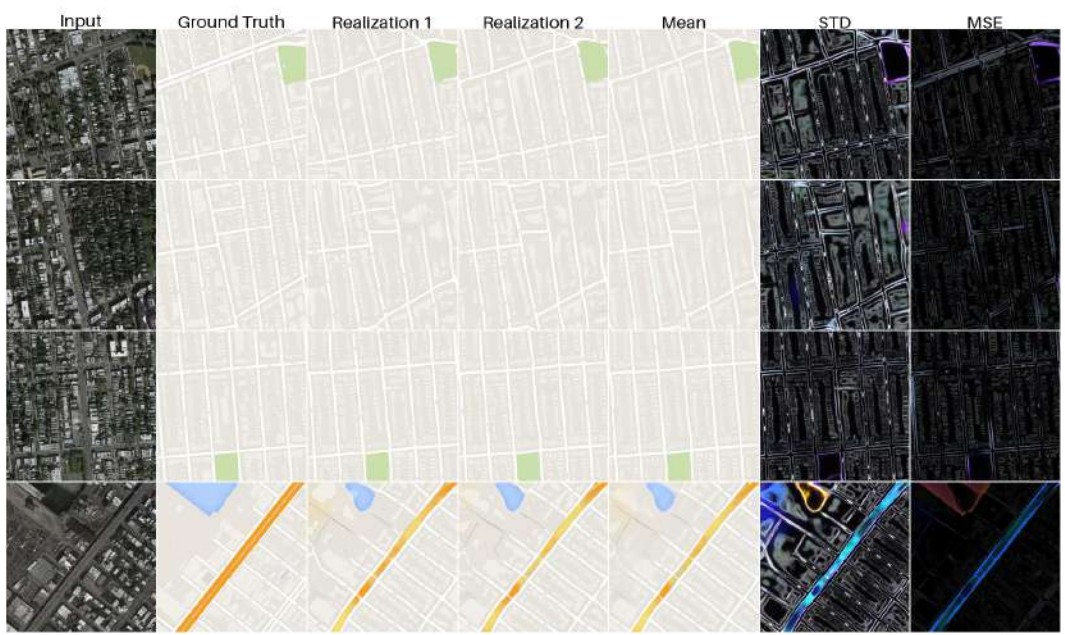

Figure 46: VAE-GAN model aerial photo realizations. From left to right: input ($y$), ground truth ($x$), 2 sample realizations, mean, STD and MSE of the 30 aerial realizations. STD is scaled by $3\times$ for visibility.

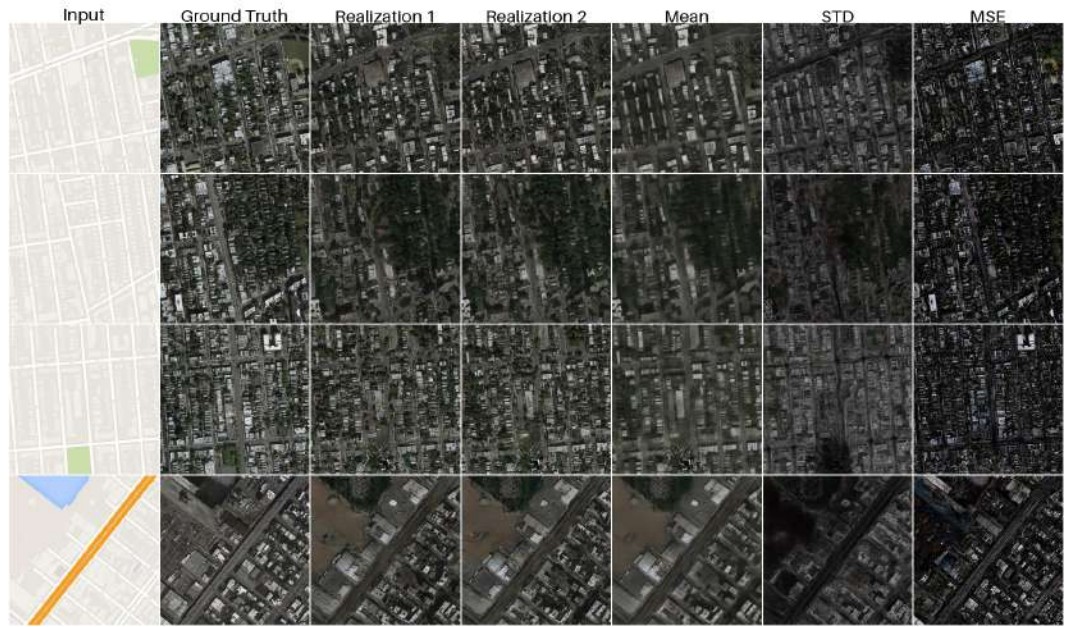

Figure 47: VAE-GAN model aerial photo realizations. From left to right: input ($y$), ground truth ($x$), 2 sample realizations, mean, STD and MSE of the 30 aerial realizations. STD is scaled by $3\times$ for visibility.

### 7.5.5 VAE-CYCLEGAN REALIZATIONS

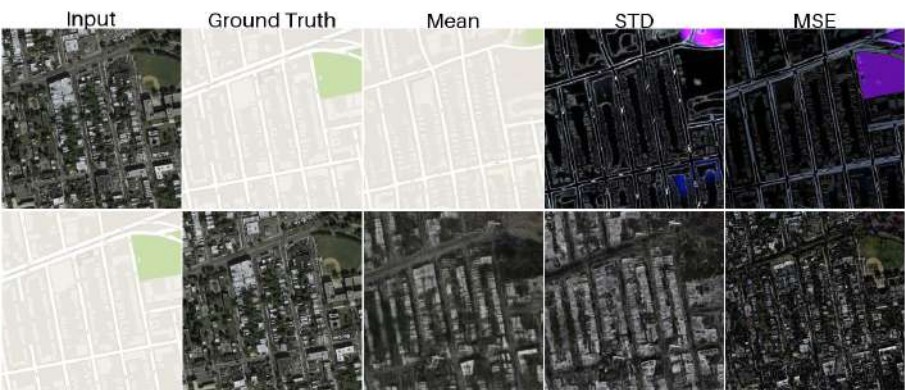

Figure 48: Mean, scaled STD (aerial $3\times$, maps $20\times$), and MSE across 30 realizations.

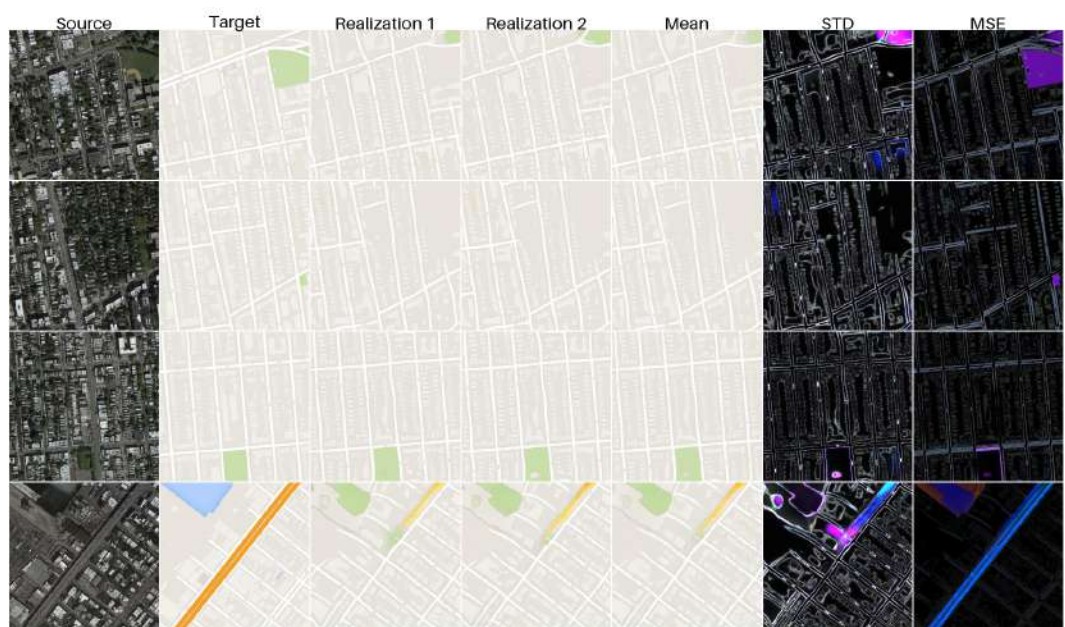

Figure 49: VAE-CycleGAN model map realizations. From left to right: input ($x$), ground truth ($y$), 2 sample realizations, mean, STD and MSE of the 30 map realizations. STD is scaled by $25\times$ for visibility.

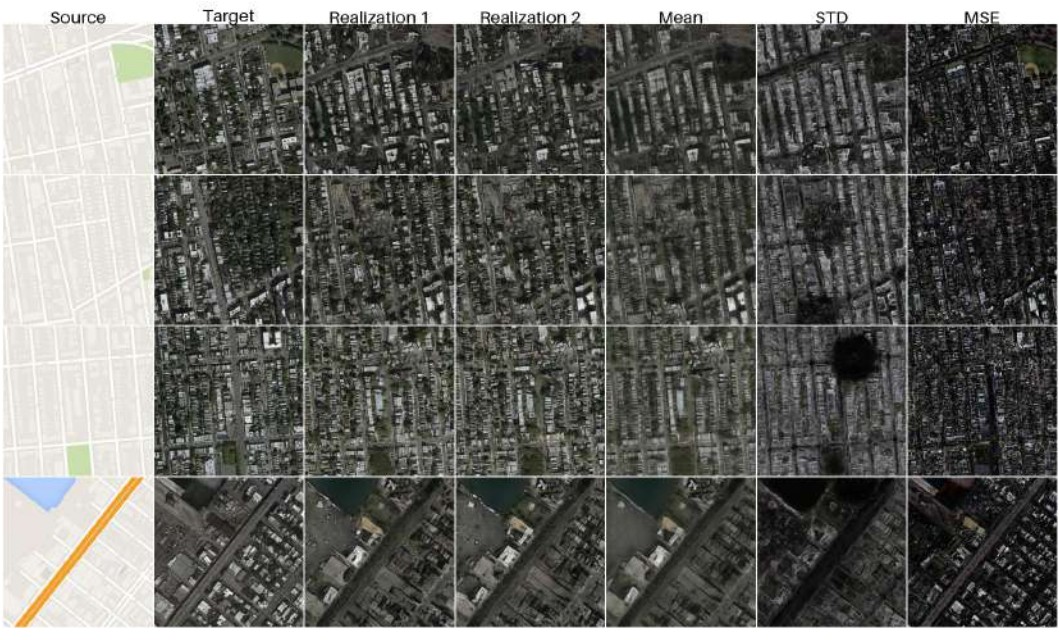

Figure 50: VAE-CycleGAN model aerial photo realizations. From left to right: input ($y$), ground truth ($x$), 2 sample realizations, mean, STD and MSE of the 30 aerial realizations. STD is scaled by $3\times$ for visibility.

## 7.6 REALIZATION METRICS

Table 9: MSE comparisons for 30 realizations, evaluated on aerial photos $(x) \leftrightarrow$ maps $(y)$. As the true translation posterior is intractable, high MSE may either indicate mean error or a (correct) region of high variance in the translation.

Table 10: VAE

| VAE | MSE $(y, \hat{y}_i)$ Maps | MSE $(x, \hat{x}_i)$ Aerial photos |
|---|---|---|
| Image 1 | 0.001517 | 0.024690 |
| Image 2 | 0.001211 | 0.023881 |
| Image 3 | 0.001258 | 0.027295 |
| Image 4 | 0.006711 | 0.023370 |
| **Average (30)** | 0.006711 | 0.023370 |

Table 11: Cycle VAE (paired)

| Cycle VAE (paired) | MSE $(y, \hat{y}_i)$ Maps | MSE $(x, \hat{x}_i)$ Aerial photos |
|---|---|---|
| Image 1 | 0.001212 | 0.023442 |
| Image 2 | 0.000959 | 0.025219 |
| Image 3 | 0.001035 | 0.027436 |
| Image 4 | 0.006002 | 0.022164 |
| **Average (30)** | 0.002302 | 0.024565 |

Table 12: Cycle VAE (unpaired)

| Cycle VAE (unpaired) | MSE $(y, \hat{y}_i)$ Maps | MSE $(x, \hat{x}_i)$ Aerial photos |
|---|---|---|
| Image 1 | 0.008954 | 0.061927 |
| Image 2 | 0.007704 | 0.056705 |
| Image 3 | 0.008576 | 0.071990 |
| Image 4 | 0.023110 | 0.053937 |
| **Average (30)** | 0.012086 | 0.061140 |

Table 13: VAE-GAN

| VAE-GAN | MSE $(y, \hat{y}_i)$ Maps | MSE $(x, \hat{x}_i)$ Aerial photos |
|---|---|---|
| Image 1 | 0.001687 | 0.028285 |
| Image 2 | 0.001156 | 0.030473 |
| Image 3 | 0.000963 | 0.031132 |
| Image 4 | 0.009378 | 0.025543 |
| **Average (30)** | 0.003296 | 0.028858 |

Table 14: VAE-CycleGAN

| VAE-CycleGAN | MSE $(y, \hat{y}_i)$ Maps | MSE $(x, \hat{x}_i)$ Aerial photos |
|---|---|---|
| Image 1 | 0.003122 | 0.033576 |
| Image 2 | 0.001444 | 0.036811 |
| Image 3 | 0.001688 | 0.036307 |
| Image 4 | 0.019514 | 0.033555 |
| **Average (30)** | 0.006442 | 0.035062 |

Input x          Output G(x)          Reconstruction F(G(x))

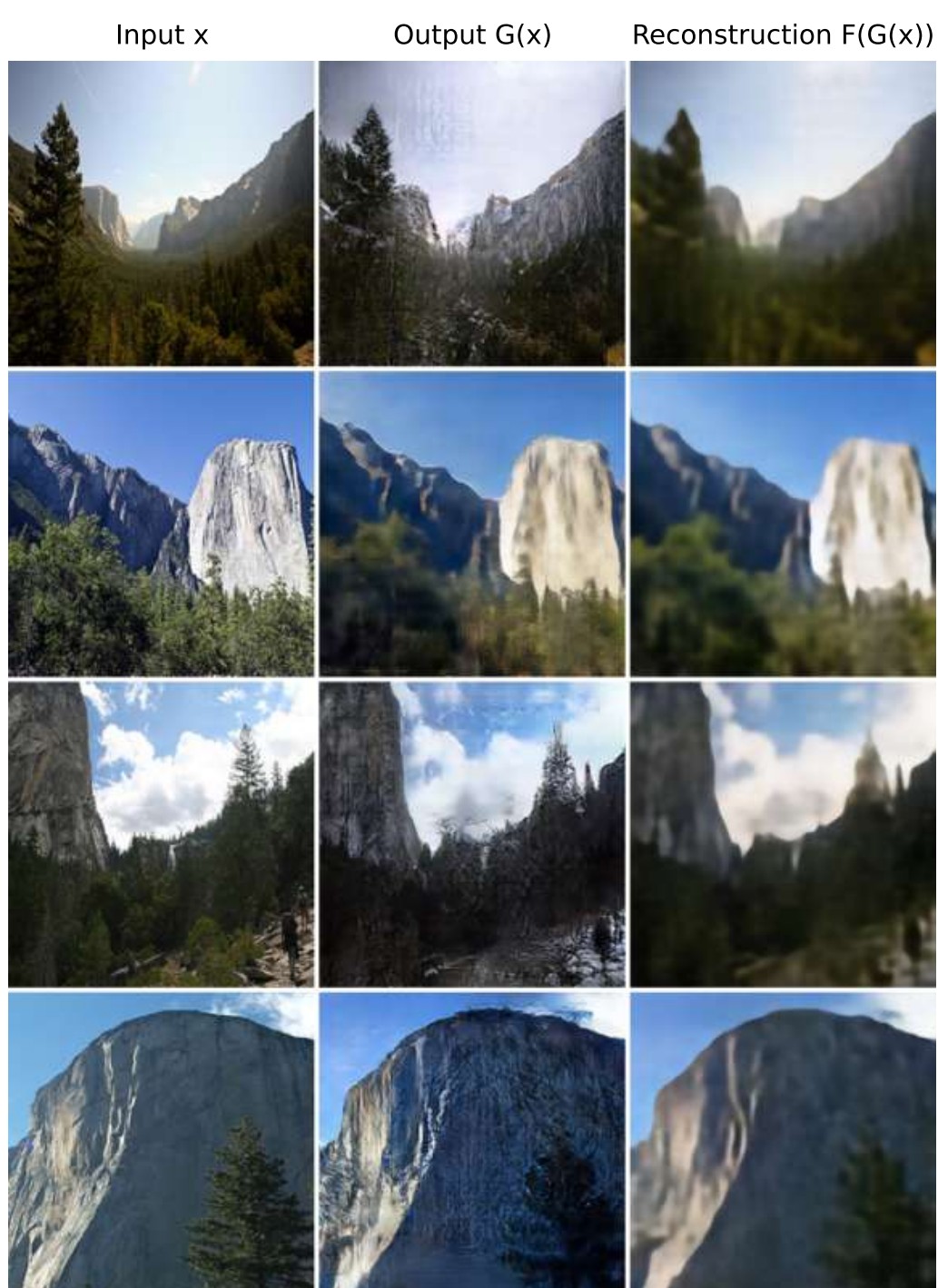

Figure 51: Summer $\leftrightarrow$ Winter : VAE-CycleGAN translated and reconstructed images, $X \rightarrow G(X) \rightarrow F(G(X))$

Input y     Output G(y)    Reconstruction F(G(y))

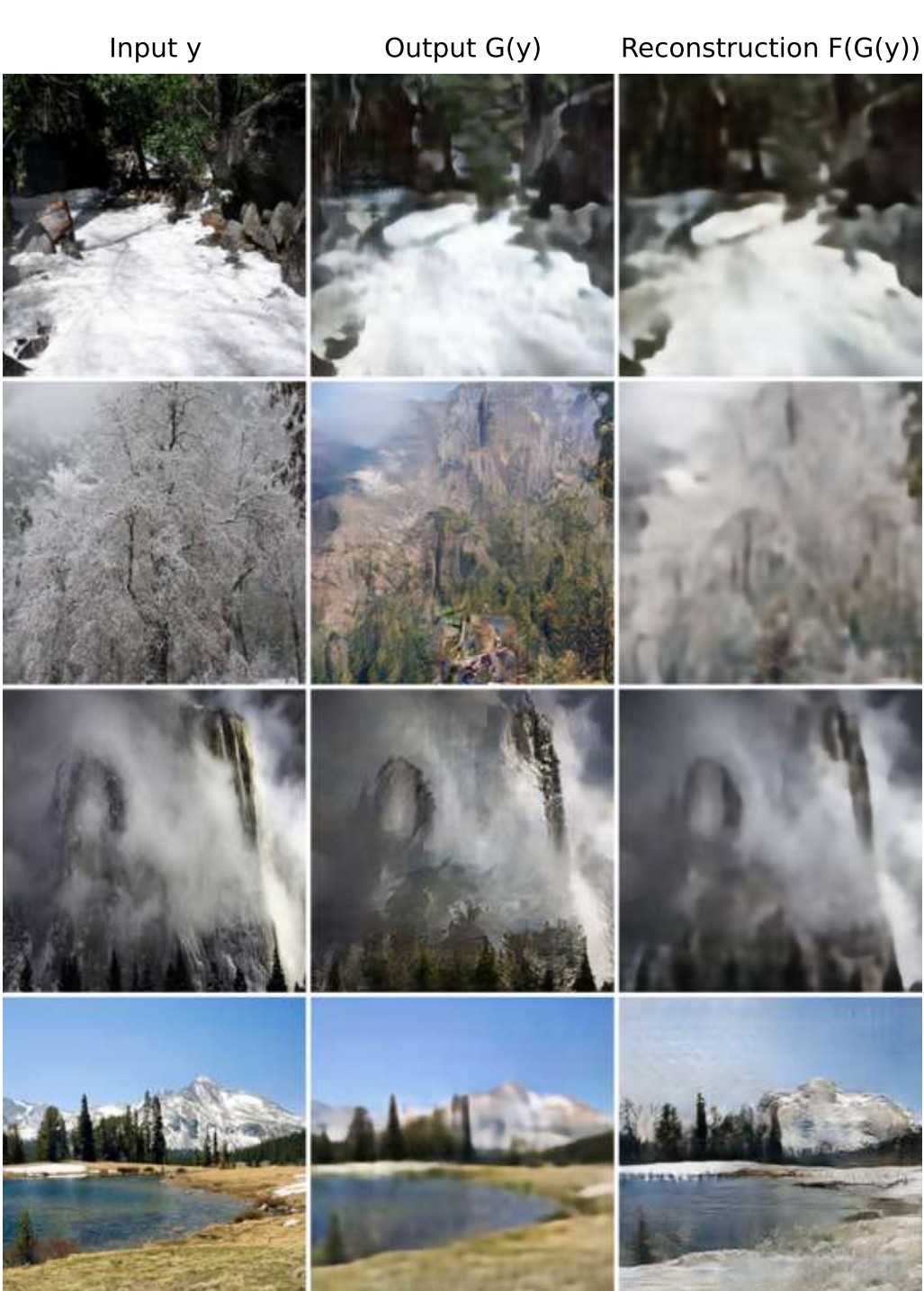

Figure 52: Summer ↔ Winter : VAE-CycleGAN translated and reconstructed images, $Y \rightarrow F(Y) \rightarrow G(F(Y))$

Input x         Output G(x)         Reconstruction F(G(x))

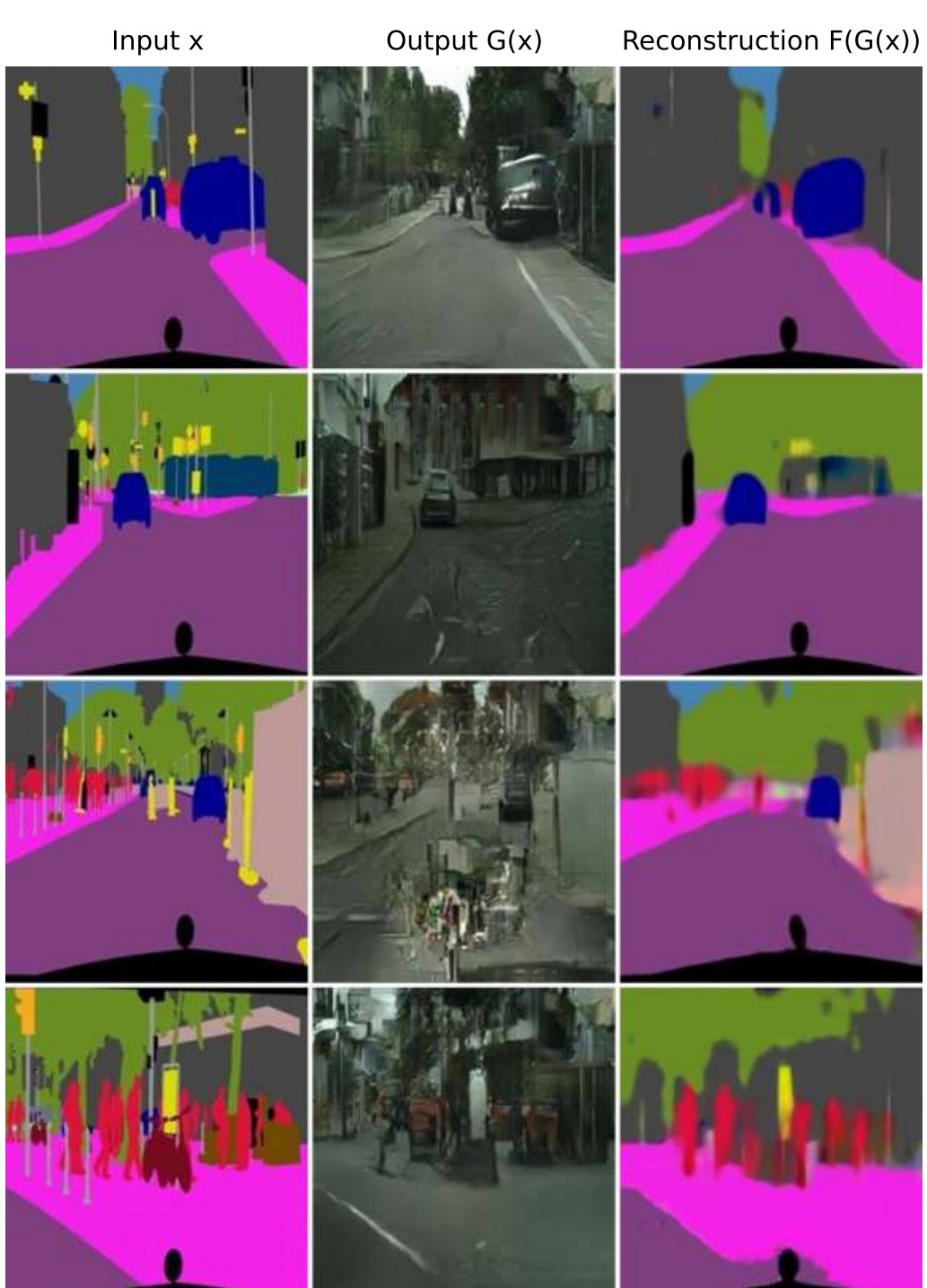

Figure 53: Label $\leftrightarrow$ Cityscape: VAE-CycleGAN translated and reconstructed images, $X \rightarrow G(X) \rightarrow F(G(X))$

Input y          Output G(y)          Reconstruction F(G(y))

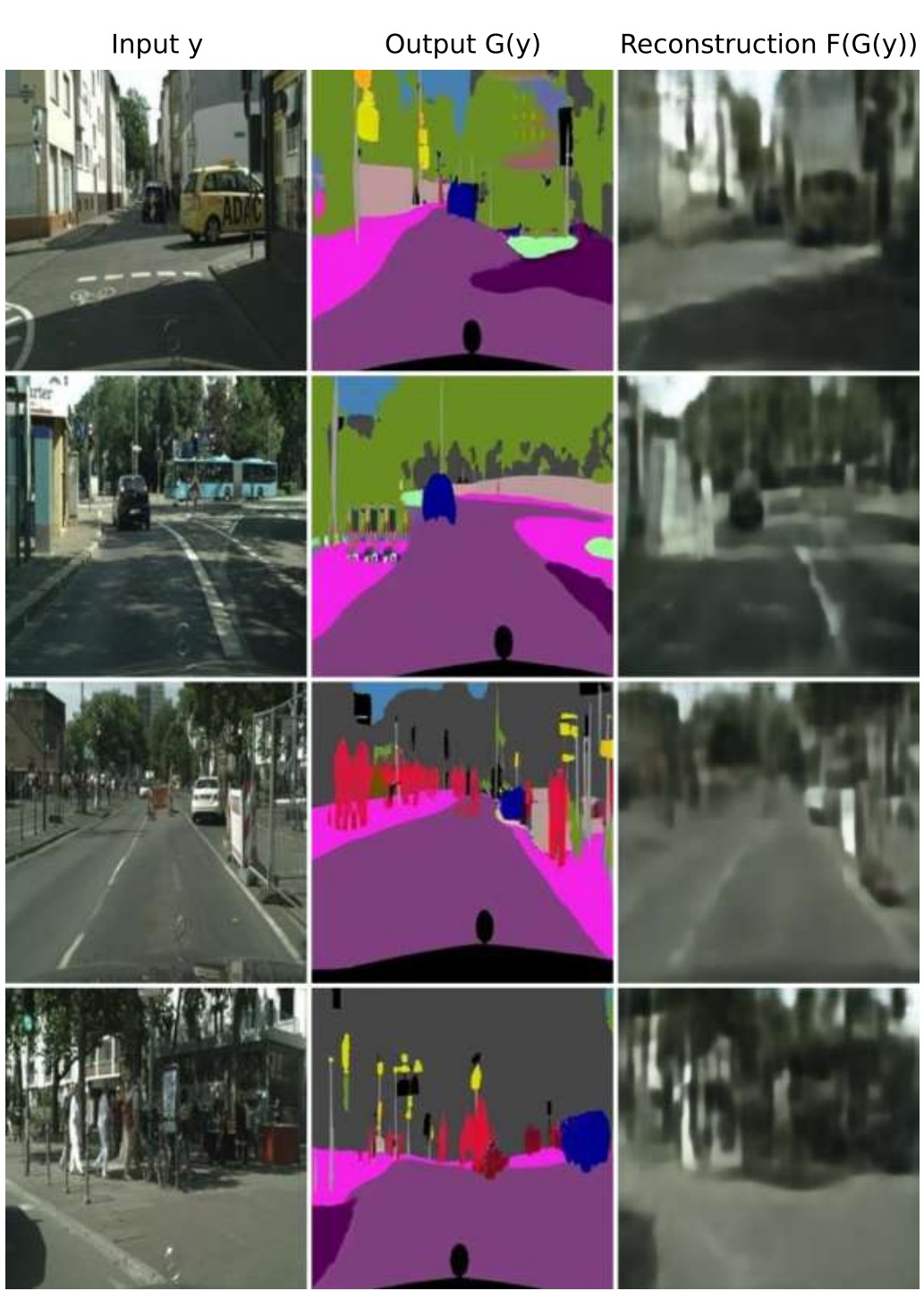

Figure 54: Label $\leftrightarrow$ Cityscape: VAE-CycleGAN translated and reconstructed images, $Y \rightarrow F(Y) \rightarrow G(F(Y))$

