# OpenReview forum: "VAE-CycleGAN: Variational Latent Representation for Unpaired Image-to-Image Translation"
_ICLR.cc/2026/Conference — Submitted to ICLR 2026_

### Official Review · Reviewer_WRrf · 2025-10-20

**Soundness:** 2
**Presentation:** 2
**Contribution:** 1
**Rating:** 0
**Confidence:** 4

**Summary:**

The paper conducts a comparison of variational and adversarial approaches to perform image translation via cycle consistency. In addition, VAE-CycleGAN is proposed, which is a hybrid VAE-GAN model that combines variational and adversarial objectives for unpaired image translation.

**Strengths:**

* It is refreshing to see a GAN/VAE paper in the era of diffusion models! The paper offers a different perspective on variational cycle consistency.
* Theoretical justification.

**Weaknesses:**

* Introduction is very short and is not entirely convincing and motivating regarding the problem the paper tackles. It also misses a lot of recent work on generative modeling and ignores the diffusion models literature.
* Figure 1: should provide a short description of the pipeline, not just “VAE-CycleGAN”.
* The qualitative results are not entirely convincing, and the visual fidelity does not seem high.
* Quantitatively, it seems that the proposed VAE-CycleGAN is outperformed by the autoencoder variants, even on generative metrics like FID. In that case, what is the significance of the proposed model? What is its advantage and why would one choose to use it?
* Section 5 is poorly written and hard to understand.
* Only one dataset.
* Overall, the manuscript reads like a “technical project report”. It is not entirely clear what the contribution is.
* Missing related work:

[1] [Jha, Ananya Harsh, et al. "Disentangling factors of variation with cycle-consistent variational auto-encoders." Proceedings of the European conference on computer vision (ECCV). 2018.](https://arxiv.org/abs/1804.10469)

[2] [Kim, Beomsu, et al. "Unpaired Image-to-Image Translation via Neural Schrödinger Bridge." The Twelfth International Conference on Learning Representations.](https://openreview.net/forum?id=uQBW7ELXfO)

[3] [Zhao, Min, et al. "Egsde: Unpaired image-to-image translation via energy-guided stochastic differential equations." Advances in Neural Information Processing Systems 35 (2022): 3609-3623.](https://arxiv.org/abs/2207.06635)

[4] [Sasaki, Hiroshi, Chris G. Willcocks, and Toby P. Breckon. "Unit-ddpm: Unpaired image translation with denoising diffusion probabilistic models." arXiv preprint arXiv:2104.05358 (2021).](https://arxiv.org/abs/2104.05358)

**Questions:**

* What is the role of $\mathcal{L}_{\text{identity}}$? It seems very counter-intuitive. Has it been ablated?
* Stability: how stable is the training of the model? How prone is it to mode collapse?
* Open-source code and reproducibility: is it the authors’ intention to publish an open-source code?
* What is the role of this kind of model compared to recent diffusion models? Why would one choose this type of model instead of a diffusion model? I would like the authors to clarify their contribution with regard to the missing related work I mentioned above.

---

> ### Author Response · Authors · 2025-12-03
> **Response to WRrf: Improved Writing, Comprehensive Literature Review, and Connection/Comparison to Diffusion**
>
> Hi Reviewer WRrf,
> Thank you so much for the detailed review.
> Your questions and comments have been very helpful for the revision.
>
> We respectfully disagree with your low opinion of our paper and hope to convince you otherwise, with the revision and responses below.
>
> Weaknesses:
> We have updated the introduction and motivation. Our apologies, we did not mean to miss the diffusion literature; it is more that we are solving a somewhat orthogonal problem. Most state-of-the-art diffusion models operate in a latent space, and several unpaired methods (like CycleDiff) require autoencoder-based translation modules. So we are improving the autoencoder for diffusion models. As such, we initially did not include them in the literature review. Please review the updated paper.
>
>
> The Figure: 1 caption has been updated, thank you.
>
>
> You are correct that the visual fidelity is not very high; we show primarily the VAE-CycleGAN, which provides stochastic realizations at significantly lower FID than the deterministic AE-CycleGAN variant. Our work also focuses on scientific accuracy (cycle-consistent ensembles) rather than visual fidelity like FID scores. Accuracy is a known weakness of diffusion models. Please also note that our AE-CycleGAN model is comparable in fidelity to other state-of-the-art models, including the ones you shared.
> We have updated the introduction to clarify, and added an FID comparison table (Table 4).
>
>
> VAE-CycleGAN will always be outperformed by autoencoder variants. The FID here does not measure the conditional distribution of translations, so does not really serve as a generative metric. We understand this is somewhat confusing and have updated the text to clarify. To our knowledge, this FID scoring is commonly used by many other papers, including UNIT-DDPM, CycleGAN and related variants, because measuring conditional FID is infeasible for sparse unpaired datasets.
>
> To be clear:
>
> The autoencoder variant is deterministic, so cannot produce distinct posterior samples, unlike the VAE. Trivially, the VAE produces a better posterior distribution as it is not collapsed to a point (given a particular sample from the prior / image to be translated). The AE posterior is a deterministic single image.
>
> The FID only implies the overall distribution of output images for the AE is better, which is expected as the AE has more accurate prediction, and thus better spans the overall feature space. This can never be beat by a variational model with the same architecture (by the perception-distortion tradeoff).
>
> Since computing the conditional FID (given a single prior image) is intractable, we resort to qualitative metrics (mean and standard deviation of realizations). As in standard Bayesian modeling, the model is considered “skilled” if the ensemble mean error is highly correlated with the standard deviation: that is, the model produces diverse outputs for regions of uncertainty. This is estimated in Section 5 on realizations, where we compare the skill of different VAE variants.
>
> We have updated the writing in Section 5; based on your questions, it seems the concept of skill and how realizations relate to skill was not clear.
>
>
> Thank you, yes. We have added two more datasets to compare FID on; performing the same extensive analysis over all VAEs/AEs does not provide more research value.
>
>
> Thank you for the opinion. We respectfully disagree with your assessment, and have made improvements to the writing. We summarize our main contributions below:
>
> CycleGAN-based translation models are deterministic; we extend it to variational cycle consistency, with a probabilistic latent space.
>
> We show that U-Net based translation is not significantly superior to autoencoder-based translation, even with the reverse diffusion process.
>
> We ablated and evaluated each and every part of the VAE-CycleGAN losses, providing a strong theoretical and experimental justification. Previous work does not completely explain nor understand why a cycle-consistency + adversarial framework is complete.
>
> The contribution of this paper is not another state-of-the-art model with marginal gains, but rather a careful framework that identifies directions for creating better autoencoders for diffusion models, in the context of unpaired translations.
> Thank you so much for these papers.We have added them to the literature review and performance comparison.

---

> ### Author Response · Authors · 2025-12-03
> **Response to WRrf: Part 2, Addressing Questions: Identity Loss, Stability, Comparison with Diffusion Models**
>
> Questions:
> Yes, identity loss has been ablated. We have added a sentence to the Problem Formulation to clarify. In general, without the identity loss the network does not stably preserve information through layers, making training from scratch unstable. This identity loss is also used in the original CycleGAN. Please also note that during fine-tuning (for the FID tests), we found that lowering the identity weight improved performance.
>
> The model is quite stable and rarely collapses; we see almost no mode collapse provided that the loss weights are not extremely unbalanced. As mentioned in the paper, we do not / have not finetuned the loss weights and other hyperparameters beyond scaling the cycle consistency to adequately preserve information.
> It is our intention to publish an open-source code, if accepted. We would like to polish the code prior to upload, which is why we have not submitted an anonymized repo.
>
> Nearly all state-of-the-art unpaired diffusion model approaches either rely on inaccurate text-based manipulation, or use a “translation module,” which is typically an autoencoder. To achieve full posterior sampling, we need semantic / latent noise, which requires a pretrained VAE of some type. So any well-performing translation model will be built on models closely matching what we describe in our paper, even if not obviously the same. It’s not so much a choice of a VAE vs a diffusion model, but rather you need a VAE to build a good diffusion model.
>
> We did investigate connecting the VAE to a diffusion model, following the DiffuseVAE paper, but could not finish training in time. Despite the architecture being exactly the same (finetuning the VAE to directly become a diffusion model, without conditioning a separate model), we already see 6+ points of FID improvement from the VAE’s ~65 FID, with only a tenth of the training. We will continue this and hope to have it ready for the camera-ready version if accepted.
>
> Please also see the response to reviewer SgCj.

---

### Official Review · Reviewer_SgCj · 2025-10-24

**Soundness:** 2
**Presentation:** 2
**Contribution:** 2
**Rating:** 2
**Confidence:** 5

**Summary:**

This work proposed VAE-CycleGAN for unpaired image-to-image translation. It combines VAE and cyclegan architecture to estimate posterior distribution for unpaired image-to-image translation task, which is a classical task for image generation. It evaluates various Autoencoder and VAE variants on satellite-to-image task. No diffusion-based methods are evaluated and compared.

**Strengths:**

- The paper provides a systematic and comprehensive comparison of different model variants (AE, VAE, AE-GAN, VAE-GAN, VAE-CycleGAN, and so on).

- Empirical results across the satellite-to-map benchmark show that VAE-CycleGAN generates high-quality translated images.

**Weaknesses:**

- missing comparison with SOTA. as diffusion models are current mainstream generative models and show unprecedent performance on image-to-image translation, why not compare with some diffusion-bsed methods? Are there any advantages over diffusion-based methods, such as Flux-Context and Qwen-ImageEdit? Motivations are needed to be further clarified.


- Insufficient evalution. this work proposeds an unpaired image-to-image translation method, but only perform evaluation on satellite to image task. I would suggest evaluating on more tasks (such as those in cyclegan paper) to justify the effectiveness of proposed method. The current experimental settings are not sufficicient to support the claim for unpaired image-to-image translation.


- writing can be improved. for example, the introduciton is too short, making it hard for readers to quickly understand. More insights and analysis are prefered in method section.

- As training process of GAN is usually unstable, how important of hyperparameter tuning in the proposed method?

**Questions:**

see above.

---

> ### Author Response · Authors · 2025-12-03
> **Response to SgCj: Comparisons to SOTA**
>
> Hi Reviewer SgCj,
>
> Please see the response to Reviewer WRrf’s first comment. There are advantages over diffusion-based methods, as we are working on an orthogonal part (VAEs) of most state-of-the-art diffusion models.
>
> Nearly all SOTA unpaired diffusion model approaches either rely on inaccurate text-based manipulation, or use a “translation module,” which is typically an autoencoder. We seek to replace this autoencoder.
>
> To the exact diffusion models you mention:
>
> FluxContext / QwenImageEdit
> For full diffusion based methods, to be consistent, unpaired translation is usually considered style transfer. We focus on scientific / engineering applications, where exact pixel-level style transfer is required (so SSIM and MSE are much more important metrics than FID).
>
> Notice for Qwen, object position is a quite fluid quantity, which is typical of diffusion models. Novel view synthesis etc. are all better diffusion tasks, but do not provide positional accuracy. You can see this example on their webpage (“image editing in avatar creator”).
> Similar examples like dress changes etc. do not change the character of the dress while retaining positional accuracy, that’s more a masking trick.
> Further, remember we do not seek to replace diffusion, just the VAE part of unpaired cycle-consistent diffusion in general.
>
> For FluxContext, they give two examples on their webpage:
> Woman next to a car: Overall we want a simple pixel-level color change, but notice the position of the woman and her shape has changed slightly (width scaling). The edit is not positionally consistent.
> An image of a couple that is restyled: The position and shapes of all the faces are now inconsistent with the original, which is acceptable for non-scientific applications, but can be catastrophic for others (examples given in paper).
>
> We have updated the introduction, motivation, and results to better clarify the connection with diffusion, and our strengths with regards to pixel-level positioning.
>
> Thank you. We have evaluated FID against other diffusion models on Label2Cityscape, Summer2Winter, and the Satellite2Map datasets.
>
> Thank you. We have significantly rewritten the introduction and parts of the results.
>
> Regarding hyperparameter tuning, please see the response to reviewer WRrf. The model is quite stable and rarely collapses; we see almost no mode collapse provided that the loss weights are not extremely unbalanced. As mentioned in the paper, we do not / have not finetuned the loss weights and other hyperparameters beyond scaling the cycle consistency to adequately preserve information.

---

### Official Review · Reviewer_CgmW · 2025-11-01

**Soundness:** 3
**Presentation:** 2
**Contribution:** 2
**Rating:** 4
**Confidence:** 4

**Summary:**

This paper considers the problem that existing deterministic models (such as CycleGAN) cannot handle ill-posed or multimodal tasks, and developed VAE-CycleGAN.

**Strengths:**

1. Integration of VAE into cycleGAN is technically solid;
2. Several analyses and visualization results are provided;
3. This paper conducts a comprehensive ablation study on 10 different AE and VAE variants.

**Weaknesses:**

1. Following the related work in the paper, integration of VAE with GAN helps to improve the generation quality. As a result, it might be straightforwad idea to introduce VAE into CycleGAN for improvement, which is the main technical contribution of the work. Therefore, the contribution and novelty of the work might not be sufficient for acceptance.
2. The experiments are performed on a single dataset (satellite to map), while most works also evaluate translation capacities on other datasets or tasks. Thus, the experimental evaluaiton is not sufficient.
3. The comparison methods are limited. For another, it seems that the proposed method does not achieve the superior performance on all the metrices, could the authors provide the corresponding explanation?
4. The organization of the work can be further improved. The Introduciton does not analyze the motivations of the work in detail.
5. Some statements are not clear,

**Questions:**

Refer to the weaknesses.

---

> ### Author Response · Authors · 2025-12-03
> **Response to Reviewer CgmW: Further Comparisons**
>
> Hi Reviewer CgmW,
>
> Thank you so much for the review. We appreciate your confidence in our work.
>
> We understand your concern that introducing a VAE into CycleGAN is relatively straightforward, however, please note that this has essentially not been done since the original CycleGAN paper. Please also see the response to Reviewer WRrf’s last question; a VAE can provide semantic noise for latent diffusion models.
>
> Thank you. We have evaluated FID against other diffusion models on Label2Cityscape, Summer2Winter, and the Satellite2Map datasets.
>
> Please see the extended response to Reviewer WRrf’s question.
> VAE-CycleGAN will always be outperformed by autoencoder variants. The FID here does not measure the conditional distribution of translations, so does not really serve as a generative metric. We understand this is somewhat confusing and have updated the text to clarify. To our knowledge, this FID scoring is commonly used by many other papers, including UNIT-DDPM, CycleGAN and related variants, because measuring conditional FID is infeasible for sparse unpaired datasets.
> By the perception-distortion tradeoff, the VAE will always have a small performance hit.
>
> Thank you. We have rewritten the introduction and improved the organization, and improved sentence clarity.

---

### Official Review · Reviewer_muvi · 2025-11-04

**Soundness:** 3
**Presentation:** 3
**Contribution:** 2
**Rating:** 2
**Confidence:** 5

**Summary:**

This paper combines the VAE, GAN and cycle consistency ideas to create a VAE-Cycle-GAN. As such, the ideas are similar to MUNIT/UNIT, and various other works that featured thereafter (e.g. VQ-VAE).

The general idea is to combine the best qualities of the VAE (latent space meaningfulness) with GANs (crisp samples), together with cycle consistency (for unpaired data). In that sense the paper succeeds in showing the effectiveness of the model.

Results are shown for standard metrics (e.g. LPIPS, FID, SSIM) to show performance on standard datasets (CityScapes and Horse-Zebra/Monet, etc.) for this type of work.

**Strengths:**

In nearly every setting, the work shows improved results over AE/VAE/GAN/CycleGAN settings. The model is able to show crispness attributable to a GAN and keeps the latent space qualities of the VAE.

The narrative generally makes sense.

**Weaknesses:**

- It is very unfortunate that I say this, but the work reads more like a recipe than an advancement. That is to say, novelty is quite limited. It might have been different if it were 2017-18.
- I am generally in agreement with the points in the paper otherwise.

**Questions:**

Have the authors considered attempting this on more modern architectures (e.g. diffusion models)?

---

> ### Author Response · Authors · 2025-12-03
> **Responses to Reviewer Muvi: Addressing Novelty Concerns and Diffusion Models**
>
> Hi Reviewer Muvi,
>
> Thank you so much for the review.
>
> Thank you. We understand that the novelty seems quite limited, especially when compared to diffusion models, but please understand that we are not proposing VAE-CycleGAN as a standalone method as much as an orthogonal complement to diffusion models. Please find a more detailed response in Reviewer WRrf’s responses.
>
> We are working on combining the VAE-CycleGAN with a DDIM style diffusion model along the DiffuseVAE framework, so that the architecture does not become a confounding factor. Unfortunately, training time was insufficient, so we hope to have it ready for the camera-ready version, if accepted. We already see 6+ points of FID improvement from the VAE’s ~65 FID, with only a tenth of the training.
>
> ----
> Just to clarify, we did not compute LPIPS or use CityScapes/Horse-Zebra/Monet in the first version of the paper. In our revision, we evaluate models against Label2Cityscapes, Maps2Satellite, and Summer2Winter.

---

### Author Response · Authors · 2025-11-21

Hi everyone,

Thank you for the detailed feedback and questions.
We are actively working on the rebuttal; our primary author was delayed by an unexpected family emergency.
We will deliver full responses by mid-next week.

In particular, we are preparing a new combined (VAECycleGAN and diffusion) test case, which we hope will address the novelty and diffusion concerns.

Thank you for your patience, and we appreciate your understanding.

---

### Meta-Review · Area_Chair_N8DL · 2025-12-24

**Summary:**

This paper proposes VAE-CycleGAN to combine VAE, GAN and cycle consistency loss to improve unpaired image-to-image translation. The reviewers agree that the paper’s main strength lie in its technically solid integration of VAE and CycleGAN, along with a systematic comparison across numerous AE/VAE/GAN variants.

Problems proposed by reviewers include:
1. The novelty of VAE-CycleGAN is limited. It is unclear what the contribution is. (muvi, CgmW, WRrf)
2. Experiments are performed only on a single dataset (satellite to map) which is insufficient. (CgmW, SgCj, WRrf)
3. Comparison methods are limited to VAEs and GANs, leaving diffusion models out of discussion. (CgmW, SgCj, WRrf)
4. Qualitative results (especially FID) do not justify the proposed model’s advantages. (CgmW, WRrf)
5. The organization of original paper is poor, including short introduction lacking motivations, unclear statements in paper and missing recent relative works. (CgmW, SgCj, WRrf)

In summary, this paper was reviewed by four experts in the field. The recommendations are Reject, Marginally Below Acceptance, Reject and Strong Reject. The reviewers had a consensus about the lack of novelty of VAE-CycleGAN and insufficient experiments. Authors’ response failed to convince them.

**Reviewer Concerns:**

**Well addressed:**
- Paper is rewritten and reorganized to be easier for understanding. Introduction are expanded to include sounder motivations. Other parts of paper are polished for smoother logic.
- Authors argue that FID only implies the overall distribution of output images from AE is better. This cannot be beat by VAE as VAE includes perception-distortion tradeoff. Other metrics are more important for this task.

**Partly addressed:**
- Experiments are performed on two more datasets (summer&winter, label&cityscape). However, only several quantitative results are provided, no qualitative results appear.


**Unsolved:**
- Authors are working on combining VAE-CycleGAN with a DDIM style diffusion model, but due to insufficient training time, no results are presented currently.
- It is still unclear where the core contribution of this paper lies, as combining VAE with GAN is already done and shows performance improvements.

**Reviewer Scores:**

**muvi (2):**

Novelty of this paper remains unclear. Reviewer is unlikely to change the score.

**CgmW (4):**

Introduction and several paragraphs are rewritten. The lowness of FID is explained by authors. However, insufficient experiments problem is not solved as only several new quantitative results are provided with no comparison methods from diffusion models. Reviewer is unlikely to change the score.

**SgCj (2):**

Introduction and several paragraphs are rewritten. However, insufficient experiments problem is not solved as only several new quantitative results are provided with no comparison methods from diffusion models. Reviewer is unlikely to change the score.

**WRrf (0):**

The role of $L_{identity}$ is ablated. The training stability is stated. However, insufficient experiments problem is not solved as only several new quantitative results are provided with no comparison methods from diffusion models. Reviewer is unlikely to change the score.

---

### Decision · Program_Chairs · 2026-01-26

Reject